# Oracle-Efficient Hybrid Online Learning with Constrained Adversaries

**Princewill Okoroafor**
Cornell University
pco9@cornell.edu

**Robert Kleinberg**
Cornell University
rdk@cs.cornell.edu

**Michael P. Kim**
Cornell University
mpk@cs.cornell.edu

## Abstract

The Hybrid Online Learning Problem, where features are drawn i.i.d. from an unknown distribution but labels are generated adversarially, is a well-motivated setting positioned between statistical and fully-adversarial online learning. Prior work has presented a dichotomy: algorithms that are statistically-optimal, but computationally intractable (Wu et al., 2023), and algorithms that are computationally-efficient (given an ERM oracle), but statistically-suboptimal (Wu et al., 2024).

This paper takes a significant step towards achieving statistical optimality and computational efficiency *simultaneously* in the Hybrid Learning setting. To do so, we consider a structured setting, where the Adversary is constrained to pick labels from an expressive, but fixed, class of functions $\mathcal{R}$. Our main result is a new learning algorithm, which runs efficiently given an ERM oracle and obtains regret scaling with the Rademacher complexity of a class derived from the Learner's hypothesis class $\mathcal{H}$ and the Adversary's label class $\mathcal{R}$. As a key corollary, we give an oracle-efficient algorithm for computing equilibria in stochastic zero-sum games when action sets may be high-dimensional but the payoff function exhibits a type of low-dimensional structure. Technically, we develop a number of novel tools for the design and analysis of our learning algorithm, including a novel Frank-Wolfe reduction with "truncated entropy regularizer" and a new tail bound for sums of "hybrid" martingale difference sequences.

## 1 Introduction

Online learning is a fundamental paradigm in machine learning, where an algorithm learns sequentially from a stream of data, making predictions and updating its model in real-time. Within the broad landscape of online learning, different assumptions can be made about how the data is generated. Two prominent extremes are the statistical setting, where data is drawn independently and identically distributed (i.i.d.) from a fixed, unknown distribution, and the fully-adversarial setting, where data is chosen by an adaptive adversary aiming to maximize the learner's error. While these are well-studied, the guarantees a learner can obtain can vary starkly between the two extremes. For example, the problem of learning thresholds from a small number of samples is straightforward in the statistical setting, but impossible in the fully-adversarial setting (Littlestone, 1988).

The Hybrid Online Learning Problem (Lazaric & Munos, 2009) has emerged as a compelling middle ground, capturing aspects of both statistical and adversarial scenarios. In this model, features are assumed to be drawn i.i.d. from an *unknown* distribution, much like in the statistical setting. The corresponding labels, however, are determined by a potentially malicious adversary. On a practical

level, this hybrid model captures real-world situations where typical instances follow statistical patterns, but the labels associated with these instances are influenced by strategic actors, system dynamics, or other worst-case forces. Theoretically, the model serves as an important frontier for exploring the limits of efficient online learning with provable guarantees.

The current state of research in Hybrid Online Learning hints at a computational-statistical divide. Algorithms that achieve statistically optimal performance (Lazaric & Munos, 2009; Wu et al., 2023) are typically computationally intractable with time and space complexity both scaling linearly in the size of the learner's hypothesis class. On the other hand, algorithms that are computationally efficient typically assume the learner has full knowledge of or unlimited sample access to the underlying feature distribution (Rakhlin et al., 2011; Haghtalab et al., 2024; Block et al., 2022), or they achieve suboptimal regret (Wu et al., 2024).

This work takes a crucial step towards bridging this gap, aiming to develop learning algorithms that are both statistically optimal and computationally efficient in the Hybrid Learning setting. To make progress on this challenging goal, we focus on a structured version of the problem. Specifically, we introduce a constraint on the adversary, assuming that the adversarial labels must be chosen from an expressive, but fixed, class of functions $\mathcal{R}$. This structural assumption allows for a more fine-grained analysis and algorithm design. Our main contribution is the development of a novel oracle-efficient learning algorithm for this structured setting.

## 1.1 Problem Formulation

We consider the following Hybrid Online Learning Problem: Let $\mathcal{X}$ be the feature space and $\mathcal{H} \subseteq [0,1]^{\mathcal{X}}$ and $\mathcal{R} \subseteq [0,1]^{\mathcal{X}}$ be the learner's hypothesis class and the adversary's constrained label function class, respectively, which are known to the learner. We assume the learner's loss function $\ell : [0,1] \times [0,1] \to \mathbb{R}$ is convex and $L$-Lipschitz with respect to its first argument for some constant $L > 0$ and measurable in the second argument. The learning process proceeds over $T$ rounds. Nature commits to a fixed, unknown distribution $\mathcal{D}$ over $\mathcal{X}$. In each round $t = 1, \ldots, T$:

1. The learner selects a hypothesis $h_t$.
2. The adversary, with knowledge of the learner's strategy but not the future feature $x_t$, selects a function $r_t$ from the adversary's label function class $\mathcal{R}$.
3. Nature samples a feature $x_t$ i.i.d. from $\mathcal{D}$. The learner incurs loss $\ell(h_t(x_t), r_t(x_t))$. The pair $(x_t, r_t)$ is revealed to the learner.

The learner's goal is to minimize its cumulative loss. The learner's strategy at time $t$ is a function of the history $(x_1, r_1), \ldots, (x_{t-1}, r_{t-1})$. We evaluate the performance of a learner by its regret with respect to the best fixed hypothesis in $\mathcal{H}$ in hindsight. The regret over $T$ rounds is defined as:

$$\text{Reg}(T) = \mathbb{E}_{x_1, \ldots, x_T \sim \mathcal{D}} \left[ \sum_{t=1}^{T} \ell(h_t(x_t), r_t(x_t)) - \min_{h \in \mathcal{H}} \sum_{t=1}^{T} \ell(h(x_t), r_t(x_t)) \right]$$

The expectation is taken over the random draws of $x_1, \ldots, x_T$ from the distribution $\mathcal{D}$. In addition to this realized regret, it will be convenient to consider the *in-expectation regret* i.e., $\sum_{t=1}^{T} \mathbb{E}_{x \sim \mathcal{D}} [\ell(h_t(x), r_t(x))] - \min_{h \in \mathcal{H}} \sum_{t=1}^{T} \mathbb{E}_{x \sim \mathcal{D}} [\ell(h(x), r_t(x))]$ Our goal is to design an oracle-efficient learner that minimizes this regret.

## 1.2 Overview of Results

Our main contribution is the development of an oracle-efficient learning algorithm for the Hybrid Online Learning Problem in a structured setting where the adversary's labeling function is constrained to a class $\mathcal{R}$. Our algorithm achieves a statistically near-optimal (up to the dependence on the adversary's constraint set $\mathcal{R}$) regret bound while being computationally efficient given access to a linear optimization oracle over the hypothesis class $\mathcal{H}$.

A key quantity characterizing the statistical complexity of function classes is the Rademacher complexity (see Section 1.4 for definition). In statistical learning theory, the Rademacher complexity of a hypothesis class $\mathcal{H}$ provides a tight characterization of the generalization error and hence the statistical error rate (Mohri et al., 2012). Our main result provides a high-probability regret bound for our hybrid learner in terms of Rademacher complexity of the function classes.

**Theorem 1.1.** Let $\mathcal{H} \subseteq [0, 1]^{\mathcal{X}}$ be a class of hypothesis functions and let $\mathcal{R} \subseteq [0, 1]^{\mathcal{X}}$ be a class of labeling functions. Let $\ell : [0, 1] \times [0, 1] \to \mathbb{R}$ be a convex, $L$-Lipschitz loss function in its first argument. There exists an online algorithm that outputs a sequence of hypothesis functions $h_1, \ldots, h_T$ such that with probability at least $1 - \delta$ over the draw of $x_1, \ldots, x_T \sim \mathcal{D}$, the following bound on the cumulative loss holds:

$$\sum_{t=1}^{T} \ell(h_t(x_t), r_t(x_t)) - \min_{h \in \mathcal{H}} \sum_{t=1}^{T} \ell(h(x_t), r_t(x_t)) \le O\left(T\mathsf{rad}_T(\ell \circ \mathcal{H} \times \mathcal{R})^1 + LT\mathsf{rad}_T(\mathcal{H}) + L\sqrt{T \log(T/\delta)}\right)$$

where $\ell \circ \mathcal{H} \times \mathcal{R}$ denotes the class of functions $\{x \mapsto \ell(h(x), r(x)) \mid h \in \mathcal{H}, r \in \mathcal{R}\}$. The algorithm runs in $O(T^2)$ time per round and makes $O(T^2)$ calls to a *linear optimization oracle* for $\mathcal{H}$ throughout $T$ rounds.

Theorem 1.1 shows that the regret of our algorithm is governed by the statistical complexity of the composite class $\ell \circ (\mathcal{H} \times \mathcal{R})$. Intuitively, this class captures the interaction between the learner's hypothesis class $\mathcal{H}$ and the adversary's labeling class $\mathcal{R}$, since each function in the class maps $x$ to the loss $\ell(h(x), r(x))$ induced by a pair $(h, r)$. Consequently, the regret scales with the Rademacher complexity of the family of losses that can arise from this interaction.

The bound is near-optimal up to its dependence on the adversary's class $\mathcal{R}$ and logarithmic factors in $T$. In particular, hybrid learning is at least as hard as the corresponding statistical learning problem, which implies a lower bound of order $LT\mathsf{rad}_T(\mathcal{H}) + L\sqrt{T \log(1/\delta)}$ on the regret (Mohri et al., 2012). Thus, even in the absence of adversarial structure, the dependence on the complexity of $\mathcal{H}$ is unavoidable.

To illustrate the bound in a concrete setting, suppose $\mathcal{H}$ is a binary-valued hypothesis class with VC dimension $d$, and the composite class $\ell \circ (\mathcal{H} \times \mathcal{R})$ is also binary-valued with VC dimension $d^*$. In this case the Rademacher complexity scales as $\mathsf{rad}_T(\mathcal{F}) = O(\sqrt{d/T})$ for VC classes (see Section 1.4), and the regret bound simplifies to $O\left(\sqrt{Td^*} + L\sqrt{Td} + L\sqrt{T \log(T/\delta)}\right)$.

Finally, the theorem highlights the role of the adversary constraint. If $\mathcal{R}$ were unrestricted (for example, $\mathcal{R} = [0, 1]^{\mathcal{X}}$), then the composite class $\ell \circ (\mathcal{H} \times \mathcal{R})$ could be arbitrarily rich, and its Rademacher complexity need not vanish with $T$. In this case, Wu et al. (2024) gives a sublinear regret bound while our theorem does not. However, in the case where $\mathcal{R}$ is constrained to be from $\mathcal{H}$, our theorem matches the lower bound from statistical learning up to log factors.

Our Hybrid Online Learning framework and our hybrid learner can be applied to the area of game theory and optimization, specifically for finding approximate solutions to stochastic saddle-point problems, or equivalently, finding approximate equilibria of stochastic zero-sum games. While it is known that oracle-efficient algorithms for finding equilibria of arbitrary zero-sum games do not exist in general (see Theorem 4 of Hazan & Koren (2016)), our results enable designing oracle-efficient algorithms whenever the game's payoff function factorizes as the composition of a bivariate convex-concave Lipschitz-continuous function with (stochastic) scalar-valued functions of each player's action. Intuitively, any such factorization of the payoff function gives the game a low-dimensional structure that is useful for efficient equilibrium computation. However, since the players' action sets themselves remain (potentially) high-dimensional, to take advantage of this low-dimensional structure in an oracle-efficient way one must design algorithms for a player to learn an approximate best-response to their opponent's adaptively-chosen action sequence in the stochastic zero-sum game, leading naturally to a Hybrid Online Learning problem.

**Corollary 1.2.** Let $\mathcal{X}$ be a domain space and $\mathcal{D}$ be a distribution over $\mathcal{X}$. Let $\mathcal{H}, \mathcal{R} \subseteq [0, 1]^{\mathcal{X}}$ be classes of functions (assumed to be closed under convex combinations) and $u : [0, 1] \times [0, 1] \to \mathbb{R}$ be a convex-concave payoff function that is $L$-Lipschitz in its first parameter. Consider the saddle-point optimization problem

$$\min_{h \in \mathcal{H}} \max_{r \in \mathcal{R}} \mathbb{E}_{x \sim \mathcal{D}}[u(h(x), r(x))]$$

Given $m$ samples from $\mathcal{D}$ and access to best-response oracles for $\mathcal{H}$ and $\mathcal{R}$, our online learning algorithm can be used to find an $\epsilon(m)$-approximate saddle point solution $(h^*, r^*)$ in polynomial time in $m$ and the complexities of $\mathcal{H}$ and $\mathcal{R}$. The approximation guarantee is $\epsilon(m) = \mathsf{rad}_m(\mathcal{F}) +$

---

[1]As defined in Section 1.4, $\mathsf{rad}_T(\mathcal{F})$ is at most 1 for any $\mathcal{F}$ and $O\left(\sqrt{d/T}\right)$ for binary classes of VC dimension $d$

$O(L \sqrt{\log m / m})$, where $\mathcal{F} = \{f : f(x) = u(h(x), r(x)) \mid h \in \mathcal{H}, r \in \mathcal{R}\}$. Note that $\mathsf{rad}_m(\mathcal{F}) \to 0$ is necessary for uniform convergence of the payoff matrix.

Finally, along the way to establishing our main result, we prove a general uniform convergence bound that may be of independent interest. This bound addresses the challenge of concentration for function classes evaluated on i.i.d. data where the functions themselves are chosen adaptively based on the previous data samples:

**Proposition 1.3.** Let $\mathcal{H}$ be a class of hypothesis functions and $\ell$ be a loss function that is $L$-Lipschitz in the first parameter. Let $x_1, x_2, \dots, x_T$ be a sequence of i.i.d samples from a fixed distribution $\mathcal{D}$. Let $r_1, r_2, \dots, r_T \in [0,1]^X$ be a sequence of functions where $r_t$ depends only on $x_1, \dots, x_{t-1}$ (and potentially prior adversarial choices). The following holds with probability at least $1 - \delta$ over the draw of $x_1, \dots, x_T$, for all $h \in \mathcal{H}$:

$$\left| \frac{1}{T} \sum_{t=1}^{T} \ell(h(x_t), r_t(x_t)) - \frac{1}{T} \sum_{t=1}^{T} \mathbb{E}_{x \sim \mathcal{D}} [\ell(h(x), r_t(x))] \right| \le O\left( L \cdot \mathsf{rad}_T(\mathcal{H}) + L \sqrt{\frac{\log(T/\delta)}{T}} \right)$$

This result provides a uniform convergence bound that effectively handles the data-dependent nature of the sequence $r_1, \dots, r_T$. The sequence $\ell(h(x_t), r_t(x_t)) - \mathbb{E}_{\mathcal{D}}[\ell(h(x), r_t(x))]$ is a martingale difference sequence since $x_t$ is sampled after the choice of $r_t$ is made. Applying Azuma-Hoeffding together with a union bound over the class $\mathcal{H}$ would only work for finite classes and would lead to a suboptimal bound of $\log |\mathcal{H}|$. We instead prove this lemma by employing a symmetrization technique and the application of a bound based on the distribution-dependent sequential Rademacher complexity, a measure introduced by Rakhlin et al. (2011). The $L$-Lipschitzness of the loss function with respect to its first parameter is key and ensures the bound depends only on the complexity of the hypothesis class $\mathcal{H}$ and the Lipschitz constant $L$, rather than the complexity of the $r_t$ sequence itself. We defer the full proof to Appendix A.2. We use Proposition 1.3 to obtain the high probability guarantee in Theorem 1.1 on the sampled sequence.

## 1.3 Technical Overview

Our technical approach begins by considering the in-expectation regret objective: to guarantee a bound on $\sum_{t=1}^{T} \mathbb{E}_{\mathcal{D}}[\ell(h_t(x), r_t(x))] - \min_{h \in \mathcal{H}} \sum_{t=1}^{T} \mathbb{E}_{\mathcal{D}}[\ell(h(x), r_t(x))]$. We note that achieving a bound on this quantity is a weaker benchmark compared to the standard regret definition (which is measured against the sum of losses on observed samples).

A key limitation in this setting is that we do not have direct access to the distribution $\mathcal{D}$. To build intuition, suppose for a moment that we had access to $m$ i.i.d. samples, $S = \{s_1, \dots, s_m\}$, from the distribution $\mathcal{D}$ *a priori*. We make the crucial observation that $m$ samples are sufficient to guarantee uniform convergence for the combined function class $\mathcal{F} = \{f : f(x) = \ell(h(x), r(x)) \; \forall h \in \mathcal{H}, r \in \mathcal{R}\}$ at a rate characterized by $\mathsf{rad}_m(\mathcal{F})$. Therefore, if we had these samples upfront, the problem could be formulated as an online learning over $\mathcal{H}$. In each round $t$, given $r_t$, the loss for a hypothesis $h$ would be the empirical average loss over the sample set $S$: $\mathbb{E}_S[\ell(h(x), r_t(x))] = \frac{1}{m} \sum_{i=1}^{m} \ell(h(s_i), r_t(s_i))$. Due to the uniform convergence property, for any $h \in \mathcal{H}$ and adaptive $r_t \in \mathcal{R}$, the empirical average $\mathbb{E}_S[\ell(h(x), r_t(x))]$ would be a good approximation of the true expectation $\mathbb{E}_{\mathcal{D}}[\ell(h(x), r_t(x))]$. Since the loss function only depends on the $m$ samples, this online learning problem is essentially an Online Convex Optimization problem with action set $(h(s_1)/m, \dots, h(s_m)/m) \in [0, 1/m]^m$ for each $h \in \mathcal{H}$ and where the loss vector in each round corresponds to the empirical losses $(\ell(h(s_1), r_t(s_1)), \dots, \ell(h(s_m), r_t(s_m))) \in [0,1]^m$. Since the action set — the projection of $\mathcal{H}$ on the $m$ samples — is a subset of $[0, 1/m]^m$ which is a subset of the $m$-dimensional simplex, then applying Follow the Regularized Leader (FTRL) achieves regret of $\sqrt{T \log m}$. Unfortunately, a naive application of FTRL will return actions on the $m$ dimensional simplex which may not correspond to any hypothesis in the class $\mathcal{H}$. To solve this problem, we introduce a Frank-Wolfe reduction to the linear optimization oracle in Section 3.

However, in the Hybrid Online Learning problem, we do not have the samples upfront. Instead, we observe samples sequentially as part of the online process itself. We thus use the dataset accumulated up to round $t - 1$, $S_t = \{x_1, \dots, x_{t-1}\}$, to define an empirical loss at round $t$: $\mathbb{E}_{S_t}[\ell(h_t(x), r_t(x))] = \frac{1}{t-1} \sum_{i=1}^{t-1} \ell(h_t(x_i), r_t(x_i))$. Unfortunately, due to the dynamically changing structure of this empirical loss function (as the dataset $D_t$ grows with $t$), this problem cannot be directly modeled as an Online Convex Optimization problem with a fixed vector space and a sequence of linear loss functions.

Despite this challenge posed by the adaptive structure of the empirical loss, we are still able to make progress by constructing an adaptive sequence of entropy regularizers. In a departure from standard FTRL analysis, the regularizers we employ are not strongly convex over the entire ambient vector space (which is of dimension $T$). This is because we never observe the "full vector" of losses or learner's actions on all $T$ samples at any given time $t < T$. Nevertheless, we bypass this difficulty by demonstrating that our adaptive entropy regularizers are strongly convex on the *relevant coordinates* (the first $t-1$ dimensions) at step $t$. This careful construction allows us to achieve a favorable bound of $O(\sqrt{T \log T})$ with respect to our in-expectation regret benchmark (the sum of expected losses).

Finally, the remaining step is to transition from the weaker benchmark (regret against the sum of expected losses over $\mathcal{D}$) to the stronger benchmark (regret against the sum of actual losses incurred on the observed samples $x_1, \ldots, x_T$). This is where uniform convergence arguments shown in Proposition 1.3 come into play, allowing us to convert the bound on the weaker benchmark into the desired bound on the standard regret definition.

## 1.4 Technical Preliminaries

**Complexity Measures**    For a function class $\mathcal{F} \subseteq \mathbb{R}^{\mathcal{X}}$ and samples $x_1, \ldots, x_T \in \mathcal{X}$, the empirical Rademacher complexity is $\widehat{\mathsf{rad}}_T(\{f|_{x_1,\ldots,x_T} : f \in \mathcal{F}\}) = \mathbb{E}_\sigma \left[ \sup_{f \in \mathcal{F}} \frac{1}{T} \sum_{t=1}^T \sigma_t f(x_t) \right]$, where $\sigma_1, \ldots, \sigma_T$ are independent random variables uniformly drawn from $\{\pm 1\}$. The Rademacher complexity at horizon $T$ with respect to distribution $\mathcal{D}$ is $\mathsf{rad}_T(\mathcal{F}) = \mathbb{E}_{x_1,\ldots,x_T \sim \mathcal{D}}[\widehat{\mathsf{rad}}_T(\{f|_{x_1,\ldots,x_T} : f \in \mathcal{F}\})]$. It is well known that for binary classes, the Rademacher complexity is tightly controlled by the VC dimension: it is both upper and lower bounded (up to logarithmic factors) by $\sqrt{\mathrm{VCdim}(\mathcal{F})/T}$ (Bartlett & Mendelson, 2003; Mohri et al., 2012). A similar result holds for real-valued classes and the fat-shattering dimension (Mohri et al., 2012).

We additionally define the composite function class: $\ell \circ \mathcal{H} \times \mathcal{R} = \{x \mapsto \ell(h(x), r(x)) \mid h \in \mathcal{H}, r \in \mathcal{R}\}$.

**Linear Optimization Oracle**    Our algorithm's computational efficiency is measured in terms of calls to a Linear Optimization Oracle for the hypothesis class $\mathcal{H}$. A Linear Optimization Oracle for $\mathcal{H}$ is an algorithm that, given a set of points $S = \{s_1, \ldots, s_m\} \subset \mathcal{X}$ and a set of weights for those points $V = \{v_1, \ldots, v_m\} \subset \mathbb{R}$, returns a hypothesis $h^* \in \mathcal{H}$ that minimizes $\sum_{i=1}^m v_i h(s_i)$ over $\mathcal{H}$. In our context, the set $S$ will typically be the set of observed samples $x_1, \ldots, x_{t-1}$ at round $t$.

## 1.5 Comparison to Prior Work

The study of hybrid online learning with an unknown i.i.d. feature source was initiated by Lazaric & Munos (2009), who obtained $O(\sqrt{dT \log T})$ regret for binary hypothesis classes with VC dimension $d$ under the absolute loss. More recently, Wu et al. (2023) extended these guarantees to real-valued hypothesis classes and general convex losses, achieving statistically near-optimal expected regret bounds for VC classes. Their algorithms rely on constructing stochastic covers of the hypothesis class, which can be computationally intractable for many natural hypothesis classes.

The first oracle-efficient algorithm for this setting was given by Wu et al. (2024), who obtained $\tilde{O}(d^{1/2}T^{3/4})$ regret for finite VC classes using a relaxation-based Follow-the-Perturbed-Leader method. While this approach is computationally efficient given an ERM oracle, the resulting regret rate is statistically suboptimal. Our work studies a structured variant of the hybrid learning problem in which the adversary is constrained to choose labels from a fixed function class $\mathcal{R}$. Under this assumption, we give an oracle-efficient algorithm whose regret scales with the Rademacher complexity of the composite class $\ell \circ (\mathcal{H} \times \mathcal{R})$, thereby recovering statistical rates whenever this composite complexity is small.

Our work is also related to recent results on smoothed and distribution-structured online learning. In particular, Block et al. (2024) study a smoothed online learning model and show that empirical risk minimization can achieve oracle-efficient error rates governed by the statistical complexity of the hypothesis class even when the base distribution is unknown. Their results imply an oracle-efficient algorithm for the realizable case of hybrid learning, in which all labels are generated by a single hypothesis from the class. A key distinction from our setting is that in the hybrid model we study, the adversary may choose a different labeling function $r_t \in \mathcal{R}$ at each round. In particular, even in

the special case $\mathcal{R} = \mathcal{H}$, the labels revealed to the learner need not be consistent with any single hypothesis $h \in \mathcal{H}$ over all rounds. Consequently, the hybrid setting we consider strictly generalizes the realizable case while still achieving regret bounds of the same statistical order.

More broadly, smoothed online learning has been studied under various assumptions on the stochastic source (Haghtalab et al., 2020; 2024; Block et al., 2022). However, many of these works assume either knowledge of the base distribution or sampling access to it. In contrast, the hybrid model considered here assumes only that features are drawn i.i.d. from an unknown distribution. Relatedly, Rakhlin et al. (2011) studied a distribution-dependent online learning framework in which Nature adaptively selects sampling distributions, but their results primarily apply to settings where the distribution is known to the learner.

Finally, our work is conceptually related to the comparative learning framework introduced by Hu & Peale (2023), in which labeling functions are also restricted to lie in a known function class. A key difference is that in comparative learning the labeling function is fixed, whereas in our hybrid setting the adversary may choose a different labeling function $r_t \in \mathcal{R}$ at each round. Understanding whether complexity measures such as the mutual VC dimension introduced by Hu & Peale (2023) can characterize the sample complexity of hybrid learning remains an interesting direction for future work.

## 2 Oracle-Efficient Hybrid Learning

In Section 2.1, we show an oracle-efficient learning algorithm for our structured hybrid setting that provides the in-expectation guarantee in Section 1.4. In Section 2.2, we prove our main result (Theorem 1.1).

### 2.1 In-Expectation Regret Guarantee using Truncated Entropy Regularization

This subsection presents and analyzes an algorithm for hybrid learning that makes use of a subroutine called an *entropy-regularized $\ell$-ERM oracle over $\mathcal{H}$*, defined as follows.

**Definition 2.1.** An entropy-regularized $\ell$-ERM oracle is initialized with a class of functions $\mathcal{H} : \mathcal{X} \rightarrow [0, 1]$. The oracle takes, as input, a subset $S \subset \mathcal{X}$ of features, a set of triples $(x_1, y_1, w_1), \ldots, (x_m, y_m, w_m) \in S \times \mathbb{R} \times \mathbb{R}$ and parameters $\eta, \epsilon$. It outputs an element $h$ in the convex hull of $\mathcal{H}$, such that $h$ minimizes (within $\varepsilon$) the function $\sum_{i=1}^{m} w_i \ell(h(x_i), y_i) + \frac{1}{\eta} \sum_{s \in S} h(s) \log(h(s) + 1)$.

We use $\log(h(x_i) + 1)$ rather than $\log h(x_i)$ in the regularizer to ensure the argument to the log is well-defined on $[0, 1]$ but more importantly, $a \log(a + 1)$ is uniformly strongly convex on the entire interval $[0, 1]$ In Section 3 below, we show how to use the Frank-Wolfe method to implement an $\varepsilon$-approximate regularized $\ell$-ERM oracle using polynomial number of calls to a linear optimization oracle.

**Theorem 2.1.** Let $\mathcal{H} \subseteq [0, 1]^{\mathcal{X}}$ be a class of hypothesis functions and let $\mathcal{R} \subseteq [0, 1]^{\mathcal{X}}$ be a class of labeling functions. Let $\ell$ be a loss function that is convex, $L$-Lipschitz in the first parameter. Given an entropy-regularized $\ell$-ERM oracle for $\mathcal{H}$, Algorithm 1 outputs a sequence of hypothesis functions $h_1, \ldots, h_T$ such that with probability at least $1 - \delta$,

$$\sum_{t=1}^{T} \mathbb{E}[\ell(h_t(x), r_t(x))] \leq \min_{h \in \mathcal{H}} \sum_{t=1}^{T} \mathbb{E}[\ell(h(x), r_t(x))] + T \cdot \text{rad}_T(\ell \circ \mathcal{H} \times \mathcal{R}) + O\left(L\sqrt{T \log T}\right)$$

The algorithm runs in time $O(T^2)$ per timestep and makes $T$ calls the entropy-regularized $\ell$-ERM oracle for $\mathcal{H}$.

**Overview of Algorithm 1** The algorithm implements a hybrid learner using the Follow The Regularized Leader (FTRL) approach over the class $\mathcal{H}$. We define a surrogate loss for each timestep based on the empirical average of the actual loss with respect to the adversary's choice $r_t$ on the samples $x_1, \ldots, x_{t-1}$ seen so far. Then we choose the approximate minimizer of the cumulative surrogate loss and an entropy regularizer that only depends on $x_1, \ldots, x_{t-1}$.

Concretely, at each timestep $t$, the algorithm outputs a predictor $h_t \in \text{conv}(\mathcal{H})$. After observing the sample $x_t$ and receiving the adversary's labeling function $r_t$, the algorithm prepares the input dataset

for the entropy-regularized ERM oracle to compute the next predictor $h_{t+1}$. The dataset provided to the oracle at step $t$ consists of triples $(x_i, y_i, w_i)$ derived from the samples $\{x_1, \ldots, x_{t-1}\}$ and the past adversarial functions $\{r_2, \ldots, r_t\}$. Specifically, for each pair of $s \in \{2, \ldots, t\}$ and $i \in \{1, \ldots, s-1\}$, the oracle receives a triple $(x_i, r_s(x_i), \frac{1}{s-1})$. The oracle finds an $\varepsilon$-approximate minimizer of this cumulative regularized empirical loss, and this minimizer becomes the predictor $h_{t+1}$ for the next round.

We introduce the following notation: let $v(h) = (h(x_1), \ldots, h(x_T))$ and $\mathcal{V} = \operatorname{conv}(\{v(h) \mid h \in \mathcal{H}\})$. We define the surrogate loss function at time $t \geq 2$ as $\tilde{\ell}_t(v) = \frac{1}{t-1} \sum_{s=1}^{t-1} \ell(v^{(s)}, r_t(x_s))$, and $\tilde{\ell}_1(v) = 0$ (where $v^{(s)}$ refers to the $s$-th coordinate of the vector $v$). Define the regularizer at time $t > 1$ as $\psi_t(v) = \frac{1}{\eta} \sum_{s=1}^{t-1} v^{(s)} \log(v^{(s)} + 1)$, and $\psi_1(v) = 0$. The algorithm at step $t$ outputs $h_t$ (corresponding to $\bar{v}_t$) where $\bar{v}_t$ is an $\varepsilon$-approximate minimizer of $F_t(v) = \sum_{s=1}^{t-1} \tilde{\ell}_s(v) + \psi_t(v)$ for $t > 1$.

---

**Algorithm 1** Hybrid Learner via FTRL with Truncated-Entropy Regularization

---

**Require:** Sequence of i.i.d. samples $\{x_t\}_{t=1}^{T} \sim \mathcal{D}^T$, time horizon $T$, failure probability $\delta$, approximation parameter $\varepsilon$ for the oracle

**Ensure:** Sequence of predictors $\{h_t\}_{t=1}^{T}$ where each $h_t \in \operatorname{conv}(\mathcal{H})$

1: Set $\eta \leftarrow \sqrt{T/L^2 \log T}, \varepsilon = L \log^{3/2} T / \sqrt{T}$
2: Initialize $h_1$ to some arbitrary hypothesis in $\mathcal{H}$
3: **for** $t = 1$ to $T$ **do**
4:     Output $h_t$, Observe $x_t$.
5:     Receive adversary function $r_t \in \mathcal{R}$.
6:     Construct the set of triples $\mathcal{S}_t = \bigcup_{s=2}^{t} \left\{ (x_i, r_s(x_i), \frac{1}{s-1}) \mid i \in \{1, \ldots, s-1\} \right\}$. (For $t = 1, \mathcal{S}_1 = \emptyset$).
7:     Obtain next predictor $h_{t+1} \in \operatorname{conv}(\mathcal{H})$ by calling the entropy-regularized ERM oracle (in Algorithm 2) with input dataset $\mathcal{S}_t$ and feature set $\{x_1, \ldots, x_t\}$:

$$h_{t+1} \leftarrow \arg\min_{h \in \operatorname{conv}(\mathcal{H})}^{\varepsilon} \left\{ \sum_{(x,y,w) \in \mathcal{S}_t} w\ell(h(x), y) + \frac{1}{\eta} \sum_{s=1}^{t} h(x_s) \log(h(x_s) + 1) \right\}.$$

8: **return:** Sequence of predictors $\{h_t\}_{t=1}^{T}$

---

**Lemma 2.2** (Approximate FTRL for Hybrid Learning). For $\eta, \varepsilon > 0$, the empirical regret of Algorithm 1 is bounded by

$$\sum_{t=1}^{T} \tilde{\ell}_t(\bar{v}_t) - \min_{u \in \mathcal{V}} \tilde{\ell}_t(u) \leq \frac{T \log 2}{\eta} + \frac{4\eta L^2 \log T}{3} + 5L\sqrt{\eta \varepsilon T}.$$

To prove this lemma, we first bound the regret of playing the exact minimizers of $F_t$. Then we appeal to the strong convexity of $F_t$ and Lipschitzness of $\ell$ to bound the loss of playing the approximate minimizers. We view this as an OCO problem with ambient vector space $\mathcal{V} \subset [0, 1]^T$ and convex loss vectors $\tilde{\ell}_t$ with an adaptive sequence of regularizers $\psi_t$. We adapt the analysis of FTRL to deal with the fact that the algorithm never observes the full vector $v \in \mathcal{V}$ and the regularizers are not strongly convex with respect to the $\ell_1$-norm of the full ambient space. However, the loss functions $\tilde{\ell}_t$ and the regularizer $\psi_t$ only depend on the first $t$ coordinates, which means its gradients are zero for coordinates $s > t$. As a result, the $\psi_t$ is strongly convex w.r.t the $\ell_1$ norm of the first $t$ coordinates and this suffices for the proof. The full proof can be found in Appendix A.1.

Now we present a uniform convergence result necessary for relating the average loss on the samples seen so far to the expected loss under the true distribution. The full proof can be found in Appendix A.1.

**Lemma 2.3.** Let $\mathcal{F} \subset [0, 1]^{\mathcal{X}}$ be a class of functions. Let $x_1, \ldots, x_T$ be a sequence of samples drawn i.i.d from a fixed distribution $\mathcal{D}$. With probability at least $1 - \delta$, for all $t \in [T], f \in \mathcal{F}$,

$$\frac{1}{t} \sum_{s=1}^{t} f(x_s) - \mathbb{E}_{x \sim \mathcal{D}}[f(x)] \leq 2\operatorname{rad}_t(\mathcal{F}) + \sqrt{\frac{\log(2T/\delta)}{t}}.$$

*Proof of Theorem 2.1.* By Lemma 2.2, the empirical regret of the sequence $\bar{v}_2, \ldots, \bar{v}_T$ (using $\tilde{\ell}_2, \ldots, \tilde{\ell}_T$) with respect to any $u \in \mathcal{V}$ is bounded by: $\sum_{t=2}^{T}(\tilde{\ell}_t(\bar{v}_t) - \tilde{\ell}_t(u)) \leq O\left(L\sqrt{T \log T}\right) + O\left(L\sqrt{\eta \varepsilon T}\right)$. Let $h \in \mathcal{H}$ be arbitrary, and $u = v(h)$. Since $\bar{v}_t = v(h_t)$,

$$\sum_{t=2}^{T}\left(\frac{1}{t-1}\sum_{s=1}^{t-1}\ell(h_t(x_s), r_t(x_s)) - \frac{1}{t-1}\sum_{s=1}^{t-1}\ell(h(x_s), r_t(x_s))\right) \leq O\left(L\sqrt{T \log T}\right) + O\left(L\sqrt{\eta \varepsilon T}\right).$$

Applying Lemma 2.3 to the function class $F = \{x \to \ell(h(x), r(x)) \; \forall h \in \mathcal{H}, r \in \mathcal{R}\}$, we have that, with probability at least $1 - \delta$, for all $t \geq 2$ and $h \in \mathcal{H}$,

$$\left|\mathbb{E}_{x \sim \mathcal{D}}[\ell(h(x), r_t(x))] - \frac{1}{t-1}\sum_{s=1}^{t-1}\ell(h(x_s), r_t(x_s))\right| \leq 2\mathsf{rad}_{t-1}(\ell \circ \mathcal{H} \times \mathcal{R}) + \sqrt{\frac{\log(2T/\delta)}{t-1}}$$

Plugging back in to the regret guarantee, we obtain that with probability at least $1 - \delta$, for all $h \in \mathcal{H}$,

$$\sum_{t=2}^{T}\mathbb{E}_{x \sim \mathcal{D}}[\ell(h_t(x), r_t(x))] - \mathbb{E}_{x \sim \mathcal{D}}[\ell(h(x), r_t(x))] \tag{1}$$

$$\leq \sum_{t=2}^{T} 2\mathsf{rad}_{t-1}(\ell \circ \mathcal{H} \times \mathcal{R}) + \sum_{t=2}^{T}\sqrt{\frac{\log(2T/\delta)}{t-1}} + O\left(L\sqrt{T \log T}\right) + O\left(L\sqrt{\eta \varepsilon T}\right) \tag{2}$$

$$\leq O\left(\sum_{t=2}^{T}\mathsf{rad}_{t-1}(\ell \circ \mathcal{H} \times \mathcal{R})\right) + O(L\sqrt{T \log T} + L\sqrt{\eta \varepsilon T}) \tag{3}$$

Using Lemma A.10, we obtain

$$\sum_{t=2}^{T}\mathsf{rad}_{t-1}(\ell \circ \mathcal{H} \times \mathcal{R}) \leq \tilde{O}(T \cdot \mathsf{rad}_T(\ell \circ \mathcal{H} \times \mathcal{R})).$$

Including the $t = 1$ term, minimizing over $h \in \mathcal{H}$ and setting $\eta = \sqrt{T/L^2 \log T}$, $\varepsilon = L\log^{3/2} T / \sqrt{T}$:

$$\sum_{t=1}^{T}\mathbb{E}[\ell(h_t(x), r_t(x))] \leq \min_{h \in \mathcal{H}}\sum_{t=1}^{T}\mathbb{E}[\ell(h(x), r_t(x))] + O(T \cdot \mathsf{rad}_T(\ell \circ \mathcal{H} \times \mathcal{R})) + O\left(L\sqrt{T \log T}\right).$$

The runtime of the algorithm is dominated by constructing the dataset $\mathcal{S}_t$ passed to the entropy-regularized ERM oracle. Since $|\mathcal{S}_t| = \sum_{s=2}^{t}(s-1) = O(t^2)$, constructing this dataset requires $O(t^2)$ time per round. $\qquad\square$

## 2.2 PROOF OF THEOREM 1.1

*Proof.* We decompose the quantity to bound: $\sum_{t=1}^{T}\ell(h_t(x_t), r_t(x_t)) - \min_{h \in \mathcal{H}}\sum_{t=1}^{T}\ell(h(x_t), r_t(x_t))] = A + B + C$ where

$$A = \sum_{t=1}^{T}(\ell(h_t(x_t), r_t(x_t)) - \mathbb{E}_{\mathcal{D}}[\ell(h_t(x), r_t(x))])$$

and

$$B = \sum_{t=1}^{T}\mathbb{E}_{\mathcal{D}}[\ell(h(x), r_t(x))] - \min_{h \in \mathcal{H}}\sum_{t=1}^{T}\mathbb{E}_{\mathcal{D}}[\ell(h(x), r_t(x))]$$

and

$$C = \min_{h \in \mathcal{H}}\sum_{t=1}^{T}\mathbb{E}_{\mathcal{D}}[\ell(h(x), r_t(x))] - \min_{h \in \mathcal{H}}\sum_{t=1}^{T}\ell(h(x_t), r_t(x_t))]$$

We bound each term with high probability and allocate a $\delta/3$ failure probability to each.

Term $A$ is a sum of martingale differences, as $Z_t = \ell(h_t(x_t), r_t(x_t)) - \mathbb{E}_{\mathcal{D}}[\ell(h_t(x), r_t(x))]$ satisfies $\mathbb{E}[Z_t|\mathcal{F}_{t-1}] = 0$. Since $\ell$ is $L$-Lipschitz over the interval $[0, 1]$, $|Z_t| \leq 2L$. By the Azuma-Hoeffding

inequality, with probability at least $1 - \delta/3$: $A \leq \sqrt{2 \sum_{t=1}^{T} L^2 \log(1/(\delta/3))} = L\sqrt{2T \log(3/\delta)} = O(L\sqrt{T \log(1/\delta)})$

Term $B$ is the in-expectation regret guarantee. By Theorem 2.1, the algorithm guarantees: $B \leq T \cdot \mathsf{rad}_T(\ell \circ \mathcal{H} \times \mathcal{R}) + O(L\sqrt{T \log T})$ with probability at least $1 - \delta/3$.

Term $C$ is the generalization gap for the best hypothesis. By Proposition 1.3, for all $h \in \mathcal{H}$, the difference between empirical and expected sums is bounded uniformly: $|\sum \ell(h, x_t, r_t) - \sum \mathbb{E}_D[\ell(h, x, r_t)]| \leq L \cdot T \cdot \mathsf{rad}_T(\mathcal{H}) + O\left(\sqrt{T \log(T/\delta)}\right)$ with probability at least $1 - \delta/3$. Using this uniform bound, we get $C \leq L \cdot T \cdot \mathsf{rad}_T(\mathcal{H}) + O(\sqrt{T \log(T/\delta)})$.

Summing the bounds for $A$, $B$, and $C$, using a union bound, we obtain that with probability at least $1 - \delta$:

$$\sum_{t=1}^{T} \ell(h_t(x_t), r_t(x_t)) - \min_{h \in \mathcal{H}} \sum_{t=1}^{T} \ell(h(x_t), r_t(x_t))]$$
$$\leq T \cdot \mathsf{rad}_T(\ell \circ \mathcal{H} \times \mathcal{R}) + L \cdot T \cdot \mathsf{rad}_T(\mathcal{H}) + O\left(L\sqrt{T \log(T/\delta)}\right)$$

This proves the regret bound. The computational efficiency follows from Theorem 2.1's use of an entropy-regularized ERM oracle, which is shown in Lemma 3.1 to be implementable efficiently using a linear optimization oracle for $\mathcal{H}$. □

## 3 Frank-Wolfe Reduction to Linear Optimization Oracle

In this section we show how to implement the entropy-regularized ERM oracle of Definition 2.1 using only access to a linear optimization oracle over $\mathcal{H}$. Our implementation follows a standard projection-free convex optimization approach based on the Frank–Wolfe method. The resulting procedure, given in Algorithm 2, computes an $\varepsilon$-approximate solution to the regularized objective using a polynomial number of calls to the linear optimization oracle for $\mathcal{H}$.

For the analysis of the Frank–Wolfe procedure we assume that the loss function $\ell$ is convex and $\beta$-smooth in its first argument. This assumption is not restrictive. If $\ell$ is convex and $L$-Lipschitz but not smooth, we can apply the standard OCO-to-OLO reduction and replace $\ell$ by its linearization at the current prediction. Concretely, if the learner predicts $a_t$ and observes label $b_t$, we use the surrogate loss $\ell'_t(a) = \nabla_1 \ell(a_t, b_t) a$, where $\nabla_1 \ell$ denotes the gradient with respect to the first argument. By convexity of $\ell$, we have $\ell(a_t, b_t) - \ell(a, b_t) \leq \nabla_1 \ell(a_t, b_t)(a_t - a)$, so any regret guarantee for the linearized losses $\ell'_t$ implies the same regret guarantee for the original losses $\ell$.

**Lemma 3.1** (Frank-Wolfe for smooth loss functions). Given a finite set of features $S \subset \mathcal{X}$, dataset $\{(x_i, y_i, w_i)\}_{i=1}^{m}$ where $x_i \in S$ for all $i$, a loss function $\ell$ that is convex and $\beta$-smooth in the first parameter, a class of functions $\mathcal{H} \subseteq [0, 1]^{\mathcal{X}}$, a linear optimization oracle for $\mathcal{H}$ over $S$, and parameters $\eta, \epsilon > 0$, Algorithm 2 returns an $\epsilon$-approximate solution $h^*$ to the entropy-regularized $\ell$-ERM problem

$$\arg\min_{h \in \mathcal{H}}^{\varepsilon} \left\{ \eta \sum_{i=1}^{m} w_i \ell(h(x_i), y_i) + \sum_{s \in S} h(s) \log(h(s) + 1) \right\}$$

after $O\left(\frac{|S|(\eta W_{\max}\beta + 1)}{\epsilon}\right)$ iterations, where $W_{\max} = \max_{s \in S} \sum_{i: x_i = s} |w_i|$ is the maximum sum of absolute weights for any feature in $S$.

**Overview of Algorithm 2:** The objective requires (approximately) solving a constrained smooth minimization problem $\min\{G(z) \mid z \in \mathcal{K}_S\}$ where $z \in [0, 1]^{|S|}$ is a vector indexed by elements of $S$, and the function $G : [0, 1]^{|S|} \to \mathbb{R}$ is defined as

$$G(z) = \eta \sum_{i=1}^{m} w_i \ell(z_{x_i}, y_i) + \sum_{s \in S} z_s \log(z_s + 1),$$

where $z_s$ denotes the component of $z$ corresponding to $s \in S$. The set $\mathcal{K}_S$ denotes the convex hull of the set of vectors $z(h) = (h(s))_{s \in S}$ as $h$ ranges over $\mathcal{H}$. In this section, we assume we are given a *linear optimization oracle* for $\mathcal{H}$ over the set $S$, that is, an algorithm for selecting the $h \in \mathcal{H}$ that minimizes

$\sum_{s \in S} c_s h(s)$ for given coefficients $\{c_s\}_{s \in S}$. Algorithm 2 below uses such an oracle to implement the Frank-Wolfe method, also known as conditional gradient descent, for approximately minimizing the convex function $G(z)$ over $\mathcal{K}_S$. At each iteration, the algorithm computes the gradient of the objective function $G(z)$ with respect to $z$, maps these components to weights $c_s$ for $s \in S$, and invokes the linear optimization oracle to find an extreme point in the original function class $\mathcal{H}$ that minimizes the corresponding linear function over $S$. After $O(\frac{|S|(\eta W_{\max}\beta+1)}{\epsilon})$ iterations, it returns an $\varepsilon$-approximate solution to the original problem.

---

**Algorithm 2** Frank-Wolfe for Entropy Regularized $\ell$-ERM

---

1: **procedure** FRANKWOLFE($\{(x_i, y_i, w_i)\}_{i=1}^m, \mathcal{H}, \eta, \varepsilon, S$)
2:     Initialize $h_1$ to an arbitrary function in $\mathcal{H}$
3:     **for** $t = 1, 2, \ldots, T$ **do**
4:         Let $z_t$ be the vector $(h_t(s))_{s \in S}$.
5:         Compute gradient components $c_s = \frac{\partial G(z_t)}{\partial z_s}$ for each $s \in S$:

$$c_s = \eta \sum_{i:x_i=s} w_i \frac{\partial \ell(h_t(s), y_i)}{\partial h} + \log(h_t(s) + 1) + \frac{h_t(s)}{h_t(s) + 1}$$

6:         Call a linear optimization oracle for $\mathcal{H}$ over $S$ with weights $\{c_s\}_{s \in S}$ to obtain $h_t' \in \mathcal{H}$
    minimizing $\sum_{s \in S} c_s h(s)$.
7:         Set $\gamma_t = \frac{2}{t+1}$
8:         Update $h_{t+1} = (1 - \gamma_t)h_t + \gamma_t h_t'$.
9:     **return** $h_{T+1}$

---

**Lemma 3.2** (Conditional Gradient Descent; (Hazan, 2023)). Let $K \subset \mathbb{R}^n$ with bounded $\ell_2$ diameter $R$. Let $f$ be a $\beta$-smooth function on $K$, then the sequence of points $x_t \in K$ computed by the conditional gradient descent algorithm satisfies

$$f(x_t) - f(x^*) \le \frac{2\beta R^2}{t + 1}$$

for all $t \ge 2$ where $x^* \in \arg\min_{x \in K} f(x)$.

The proof of Lemma 3.1 uses the formulation of the problem as minimizing a smooth convex function $G(z)$ over a bounded set $\mathcal{K}_S \subset \mathbb{R}^{|S|}$. We then compute and bound the smoothness constant of $G(z)$ and the diameter of $\mathcal{K}_S$. Finally, we apply the standard convergence guarantee for the Frank-Wolfe algorithm in Lemma 3.2 to obtain the stated convergence guarantee.

## 4 APPLICATION TO GAMES

**Corollary 4.1.** Let $X$ be a domain space and $\mathcal{D}$ be a distribution over $X$. Let $\mathcal{H}, \mathcal{R} \subseteq [0, 1]^X$ be classes of functions (assumed to be closed under convex combinations) and $u : [0, 1] \times [0, 1] \to \mathbb{R}$ be a convex-concave payoff function that is $L$-Lipschitz in its first parameter. Consider the saddle-point optimization problem

$$\min_{h \in \mathcal{H}} \max_{r \in \mathcal{R}} \mathbb{E}_{x \sim \mathcal{D}}[u(h(x), r(x))]$$

Given $m$ samples from $\mathcal{D}$ and access to best-response oracles for $\mathcal{H}$ and $\mathcal{R}$, our online learning algorithm can be used to find an $\epsilon(m)$-approximate saddle point solution $(h^*, r^*)$ in polynomial time in $m$ and the complexities of $\mathcal{H}$ and $\mathcal{R}$. The approximation guarantee is $\epsilon(m) = \mathsf{rad}_m(\mathcal{F}) + O(L\sqrt{\log m/m})$, where $\mathcal{F} = \{f : f(x) = u(h(x), r(x)) \mid h \in \mathcal{H}, r \in \mathcal{R}\}$. Note that $\mathsf{rad}_m(\mathcal{F}) \to 0$ is necessary for uniform convergence of the payoff matrix.

To prove Corollary 4.1, we feed the $m$ samples from $\mathcal{D}$ sequentially into our hybrid learner. For each timestep $t$, we choose $r_t$ to be the best response function to the $h_t$ the algorithm outputs. We use the same $m$ samples to compute the best response function $r_t = \text{argmax}_{r \in \mathcal{R}} \sum_{i=1}^m u(h_t(x_i), r(x_i))$ and this will be close to the true best response due to uniform convergence. Finally, using standard minimax analysis, we argue in Appendix C that the process converges to an approximate equilibrium.

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
