# A   Deferred Proof from Section 2

## A.1   Deferred Proofs from Section 2.1

### A.1.1   Reference Lemmas

**Lemma A.1** (Lemma 7.8 of Orabona (2023))**.** Assume $V$ is convex. If $F_t$ is closed, subdifferentiable, and strongly convex in $V$, then $v_t$ exists and is unique. In addition, assume $\partial \tilde{\ell}_t(v_t)$ to be non-empty and $F_t + \tilde{\ell}_t$ to be closed, subdifferentiable, and $\lambda_t$-strongly convex w.r.t. $\|\cdot\|$ in $V$. Then, we have

$$F_t(v_t) - F_{t+1}(v_{t+1}) + \ell_t(v_t) \leq \frac{\|g_t\|_\star^2}{2\lambda_t} + \psi_t(v_{t+1}) - \psi_{t+1}(v_{t+1}), \forall g_t \in \partial \tilde{\ell}_t(v_t).$$

**Theorem A.2** (Theorem 2.18 of Orabona (2023))**.** Let $f_1, \ldots, f_m$ be proper functions on $\mathbb{R}^d$, and $f = f_1 + \cdots + f_m$. Then, $\partial f(v) \supseteq \partial f_1(v) + \cdots + \partial f_m(v), \forall v$. Moreover, if $f_1, \ldots, f_m$ are also convex, closed, and $\mathrm{dom}\, f_m \cap \bigcap_{i=1}^{m-1} \mathrm{int}\, \mathrm{dom}\, f_i \neq \emptyset$, then actually $\partial f(v) = \partial f_1(v) + \cdots + \partial f_m(v), \forall v$.

**Theorem A.3** (Theorem 3.3 of Mohri et al. (2012))**.** Fix distribution $D|_X$ and parameter $\delta \in (0, 1)$. If $\mathcal{F} \subseteq \{f : X \rightarrow [-1, 1]\}$ and $S = \{x_1, \ldots, x_m\}$ is drawn i.i.d. from $D|_X$, then with probability $\geq 1 - \delta$ over the draw of $S$, for every function $f \in \mathcal{F}$,

$$\mathbb{E}_D[f(x)] \leq \mathbb{E}_S[f(x)] + 2\mathrm{rad}_m(\mathcal{F}) + \sqrt{\frac{\ln(1/\delta)}{m}}. \tag{1}$$

In addition, with probability $\geq 1 - \delta$, for every function $f \in \mathcal{F}$,

$$\mathbb{E}_D[f(x)] \leq \mathbb{E}_S[f(x)] + 2\hat{\mathrm{rad}}_m(\mathcal{F}) + 3\sqrt{\frac{\ln(2/\delta)}{m}}. \tag{2}$$

### A.1.2   Proof of Lemma 2.3

*Proof.* For any fixed $t \in [T]$, consider the set of the first $t$ samples $S_t = \{x_1, \ldots, x_t\}$, drawn i.i.d. from $D$. The class $\mathcal{F} \subset \{X \rightarrow [0, 1]\} \subseteq \{X \rightarrow [-1, 1]\}$. Also, consider the class $-\mathcal{F} = \{-f \mid f \in \mathcal{F}\}$, which is also a subset of $\{X \rightarrow [-1, 1]\}$, and $\mathrm{rad}_t(-\mathcal{F}) = \mathrm{rad}_t(\mathcal{F})$.

For a fixed $t$, applying Theorem A.3 to $\mathcal{F}$ with confidence $\delta_t/2$, we get that with probability at least $1 - \delta_t/2$, for all $f \in \mathcal{F}$:

$$\mathbb{E}_D[f(x)] \leq \mathbb{E}_{S_t}[f(x)] + 2\mathrm{rad}_t(\mathcal{F}) + \sqrt{\frac{\ln(2/\delta_t)}{t}}$$

This provides an upper bound on $\mathbb{E}_{S_t}[f(x)] - \mathbb{E}_D[f(x)]$.

Applying Theorem A.3 to $-\mathcal{F}$ with confidence $\delta_t/2$, we get that with probability at least $1 - \delta_t/2$, for all $g \in -\mathcal{F}$:

$$\mathbb{E}_D[g(x)] \leq \mathbb{E}_{S_t}[g(x)] + 2\mathrm{rad}_t(-\mathcal{F}) + \sqrt{\frac{\ln(2/\delta_t)}{t}}$$

Substituting $g = -f$ for $f \in \mathcal{F}$ and using $\mathrm{rad}_t(-\mathcal{F}) = \mathrm{rad}_t(\mathcal{F})$:

$$-\mathbb{E}_D[f(x)] \leq -\mathbb{E}_{S_t}[f(x)] + 2\mathrm{rad}_t(\mathcal{F}) + \sqrt{\frac{\ln(2/\delta_t)}{t}}$$

Multiplying by -1, we get a lower bound on $\mathbb{E}_{S_t}[f(x)] - \mathbb{E}_D[f(x)]$:

$$\mathbb{E}_D[f(x)] \geq \mathbb{E}_{S_t}[f(x)] - \left(2\mathrm{rad}_t(\mathcal{F}) + \sqrt{\frac{\ln(2/\delta_t)}{t}}\right)$$

Combining the upper and lower bounds, with probability at least $1 - \delta_t/2 - \delta_t/2 = 1 - \delta_t$, for all $f \in \mathcal{F}$:

$$\left|\frac{1}{t}\sum_{s=1}^t f(x_s) - \mathbb{E}_D[f(x)]\right| \leq 2\mathrm{rad}_t(\mathcal{F}) + \sqrt{\frac{\ln(2/\delta_t)}{t}}$$

Finally, we apply a union bound over $t \in [T]$. We want the bound to hold for all $t \in [T]$ with overall probability at least $1 - \delta$. Let $F_t$ be the event that the inequality above does not hold for a specific $t$. We have $P(F_t) \le \delta_t$. By the union bound, $P(\cup_{t=1}^{T} F_t) \le \sum_{t=1}^{T} P(F_t) \le \sum_{t=1}^{T} \delta_t$. Setting $\delta_t = \delta/T$, we get $\sum_{t=1}^{T} \delta/T = \delta$.

Thus, with probability at least $1 - \delta$, for all $t \in [T]$ and for all $f \in \mathcal{F}$:

$$\left| \frac{1}{t} \sum_{s=1}^{t} f(x_s) - \mathbb{E}_D[f(x)] \right| \le 2\mathsf{rad}_t(\mathcal{F}) + \sqrt{\frac{\ln(2/(\delta/T))}{t}} = 2\mathsf{rad}_t(\mathcal{F}) + \sqrt{\frac{\ln(2T/\delta)}{t}}$$

$\square$

### A.1.3 Proof of Lemma 2.2

Before presenting the proof, we first state the following key lemma that analyzes the regret of the hybrid learner if the algorithm used an exact ERM instead of an approximate one.

**Lemma A.4** (Exact FTRL for Hybrid Learning). Consider the (exact minimizer) Follow-the-Regularized-Leader (FTRL) approach with regularizer $\psi_t(v) = \frac{1}{\eta} \sum_{s=1}^{t-1} v^{(s)} \log(v^{(s)} + 1)$ and loss functions $\tilde{\ell}_t(v) = \frac{1}{t-1} \sum_{s=1}^{t-1} \ell(v^{(s)}, r_t(x_s))$ (for $t > 1$), where at each time step $t$, the decision $v_t \in V \subseteq [0, 1]^d$ minimizes $F_t(v) = \psi_t(v) + \sum_{i=1}^{t-1} \tilde{\ell}_i(v)$. Then, for any $u \in V$, the empirical regret is bounded by:

$$\sum_{t=1}^{T} (\tilde{\ell}_t(v_t) - \tilde{\ell}_t(u)) \le \frac{T \log 2}{\eta} + \frac{2\eta L^2}{3}(1 + \log(T - 1)).$$

If $\eta = \sqrt{T/L^2 \log T}$, the empirical regret is bounded by $O\left(L \sqrt{T \log T}\right)$.

The proof of this lemma has been deferred to later in the section. We now present the proof of Lemma 2.2.

*Proof of Lemma 2.2.* We want to bound the empirical regret of the approximate FTRL algorithm, $\sum_{t=1}^{T} (\tilde{\ell}_t(\bar{v}_t) - \tilde{\ell}_t(u))$ for any $u \in V$. We decompose the sum as:

$$\sum_{t=1}^{T} (\tilde{\ell}_t(\bar{v}_t) - \tilde{\ell}_t(u)) = \sum_{t=1}^{T} (\tilde{\ell}_t(\bar{v}_t) - \tilde{\ell}_t(v_t)) + \sum_{t=1}^{T} (\tilde{\ell}_t(v_t) - \tilde{\ell}_t(u)),$$

where $v_t \in V$ is the exact minimizer of $F_t(v)$ at time $t$. The second term on the right-hand side is the empirical regret of the exact FTRL algorithm. By Lemma A.4, this term is bounded by:

$$\sum_{t=1}^{T} (\tilde{\ell}_t(v_t) - \tilde{\ell}_t(u)) \le O\left(L \sqrt{T \log T}\right)$$

Now, we bound the first term, which is the accumulated difference in loss between the approximate and exact minimizers: $\sum_{t=1}^{T} (\tilde{\ell}_t(\bar{v}_t) - \tilde{\ell}_t(v_t))$. Since $\bar{v}_t$ is an $\varepsilon$-approximate minimizer of $F_t(v)$, we have $F_t(\bar{v}_t) \le F_t(v_t) + \varepsilon$. Using the strong convexity of $F_t$ with parameter $\lambda_t = \frac{3}{4\eta(t-1)}$ for $t > 1$ with respect to the $\ell_1$ norm of the first $t - 1$ coordinates, the difference in function values is related to the squared distance between the points:

$$F_t(\bar{v}_t) - F_t(v_t) \ge \frac{\lambda_t}{2} \|\bar{v}_t^{(1:t-1)} - v_t^{(1:t-1)}\|_1^2.$$

Here the superscript refers to the first $t - 1$ coordinates of the vector $v_t$. Combining with the $\varepsilon$-optimality, we get a bound on the $\ell_1$ distance between $\bar{v}_t^{(1:t-1)}$ and $v_t^{(1:t-1)}$:

$$\frac{\lambda_t}{2} \|\bar{v}_t^{(1:t-1)} - v_t^{(1:t-1)}\|_1^2 \le \varepsilon \implies \|\bar{v}_t^{(1:t-1)} - v_t^{(1:t-1)}\|_1 \le \sqrt{\frac{2\varepsilon}{\lambda_t}}.$$

The function $\tilde{\ell}_t(v)$ is convex and L-Lipschitz. For $t > 1$, the $\ell_\infty$ norm of its subgradient is bounded by $\|\partial \tilde{\ell}_t(v)\|_\infty \le \frac{L}{t-1}$. The difference in loss can be bounded using the Lipschitz property:

$$\tilde{\ell}_t(\bar{v}_t) - \tilde{\ell}_t(v_t) \le \|\partial \tilde{\ell}_t\|_\infty \|\bar{v}_t^{(1:t-1)} - v_t^{(1:t-1)}\|_1 \le \frac{L}{t-1} \|\bar{v}_t^{(1:t-1)} - v_t^{(1:t-1)}\|_1, \quad \text{for } t > 1.$$

Substitute the bound on $\|\bar{v}_t^{(1:t-1)} - v_t^{(1:t-1)}\|_1$:

$$\tilde{\ell}_t(\bar{v}_t) - \tilde{\ell}_t(v_t) \le \frac{L}{t-1} \sqrt{\frac{2\varepsilon}{\lambda_t}}, \quad \text{for } t > 1.$$

Using $\lambda_t = \frac{3}{4\eta(t-1)}$ for $t > 1$:

$$\tilde{\ell}_t(\bar{v}_t) - \tilde{\ell}_t(v_t) \le \frac{L}{t-1} \sqrt{\frac{2\varepsilon}{\frac{3}{4\eta(t-1)}}} = \frac{L}{t-1} \sqrt{\frac{8\eta\varepsilon(t-1)}{3}} = L\sqrt{\frac{8\eta\varepsilon}{3(t-1)}}, \quad \text{for } t > 1.$$

Now, sum this bound from $t = 2$ to $T$

$$\sum_{t=1}^{T}(\tilde{\ell}_t(\bar{v}_t) - \tilde{\ell}_t(v_t)) \le \sum_{t=2}^{T} L\sqrt{\frac{8\eta\varepsilon}{3(t-1)}} = L\sqrt{\frac{8\eta\varepsilon}{3}} \sum_{t=2}^{T} \frac{1}{\sqrt{t-1}}.$$

We use the bound $\sum_{k=1}^{T-1} \frac{1}{\sqrt{k}} \le 1 + \int_1^{T-1} x^{-1/2}dx = 1 + [2\sqrt{x}]_1^{T-1} = O(\sqrt{T})$.

$$\sum_{t=1}^{T}(\tilde{\ell}_t(\bar{v}_t) - \tilde{\ell}_t(v_t)) \le L\sqrt{\frac{8\eta\varepsilon}{3}} O(\sqrt{T}) = O(L\sqrt{\eta\varepsilon T}).$$

Combining the bounds for the two terms in the regret decomposition:

$$\sum_{t=1}^{T}(\tilde{\ell}_t(\bar{v}_t) - \tilde{\ell}_t(u)) \le \left(\frac{T\log 2}{\eta} + \frac{2\eta L^2}{3}(1 + \log(T-1))\right) + O(L\sqrt{\eta\varepsilon T}).$$

$\square$

To prove Lemma A.4, we first present the following helper lemmas:

**Lemma A.5** (Lemma 7.1 of Orabona (2023))**.** The Follow-the-Regularized-Leader (FTRL) algorithm, at each time step $t$ from 1 to $T$, outputs a decision $v_t$ that minimizes $F_t(v)$ over a closed and non-empty set $V \subseteq \mathbb{R}^d$, where $F_t(v) = \psi_t(v) + \sum_{i=1}^{t-1} \tilde{\ell}_i(v)$. That is,

$$v_t \in \text{argmin}_{v \in V} F_t(v).$$

Assume that $\text{argmin}_{v \in V} F_t(v)$ is non-empty, and let $v_t$ be an element of this set. Then, for any $u \in \mathbb{R}^d$, we have:

$$\sum_{t=1}^{T}(\tilde{\ell}_t(v_t) - \tilde{\ell}_t(u)) = \psi_{T+1}(u) - \min_{v \in V} \psi_1(v) + \sum_{t=1}^{T}[F_t(v_t) - F_{t+1}(v_{t+1}) + \tilde{\ell}_t(v_t)] + (F_{T+1}(v_{T+1}) - F_{T+1}(u)).$$

The proof of this lemma can be found in the referenced text.

**Lemma A.6** (Variant of Lemma 7.8 of Orabona (2023))**.** Assume $V$ is convex and $\partial\tilde{\ell}_t(v_t)$ is non-empty for the Follow-the-Regularized-Leader (FTRL) approach with regularizer $\psi_t(v) = \frac{1}{\eta} \sum_{s=1}^{t-1} v^{(s)} \log(v^{(s)} + 1)$ and loss functions $\tilde{\ell}_t(v) = \frac{1}{t-1} \sum_{s=1}^{t-1} \ell(v^{(s)}, r_t(x_s))$ (for $t > 1$). Then for $F_t(v) = \psi_t(v) + \sum_{i=1}^{t-1} \tilde{\ell}_i(v)$, it holds that for any $g_t \in \partial\tilde{\ell}_t(v_t)$, we have

$$F_t(v_t) - F_{t+1}(v_{t+1}) + \tilde{\ell}_t(v_t) \le \frac{\|g_t\|_\infty^2}{2\lambda_t} + \psi_t(v_{t+1}) - \psi_{t+1}(v_{t+1}).$$

for $\lambda_t = \frac{3}{4\eta(t-1)}$

Note that a direct application of Lemma A.1 (Lemma 7.8 of Orabona (2023)) can be challenging because the objective function $F_t$ depends only on the first $t - 1$ coordinates of $v$, while the domain $V$ is in a higher-dimensional space. The following lemma provides a bound on the change in the objective function plus current loss, similar to Lemma A.1, adapted for this structure. The proof is deferred to later in the section.

**Lemma A.7.** For $t > 1$, the regularizer $\psi_t(v) = \frac{1}{\eta} \sum_{s=1}^{t-1} v^{(s)} \log(v^{(s)} + 1)$ defined over $V^{(t-1)} \subseteq [0,1]^{t-1}$ is $\frac{3}{4\eta(t-1)}$-strongly convex with respect to the $\ell_1$ norm.

The proof of the lemma has been deferred to later in this section.

*Proof of Lemma A.4.* From Lemma A.5, for any $u \in V$, we have:

$$\sum_{t=1}^{T}(\tilde{\ell}_t(v_t) - \tilde{\ell}_t(u)) = \psi_{T+1}(u) - \min_{v \in V} \psi_1(v) + \sum_{t=1}^{T}[F_t(v_t) - F_{t+1}(v_{t+1}) + \tilde{\ell}_t(v_t)] + (F_{T+1}(v_{T+1}) - F_{T+1}(u)).$$

From the definition of $\psi_1$, $\psi_1(v) = \frac{1}{\eta}\sum_{s=1}^{0} v^{(s)} \log(v^{(s)} + 1) = 0$. Thus, $\min_{v \in V} \psi_1(v) = 0$. The equality becomes:

$$\sum_{t=1}^{T}(\tilde{\ell}_t(v_t) - \tilde{\ell}_t(u)) = \psi_{T+1}(u) + \sum_{t=1}^{T}[F_t(v_t) - F_{t+1}(v_{t+1}) + \tilde{\ell}_t(v_t)] + (F_{T+1}(v_{T+1}) - F_{T+1}(u)).$$

By Lemma A.6, for $t > 1$, we have for $\lambda_t = \frac{3}{4\eta(t-1)}$:

$$F_t(v_t) - F_{t+1}(v_{t+1}) + \tilde{\ell}_t(v_t) \leq \frac{\|g_t\|_\infty^2}{2\lambda_t} + \psi_t(v_t) - \psi_{t+1}(v_{t+1}), \quad \forall g_t \in \partial\tilde{\ell}_t(v_t).$$

Summing this inequality from $t = 1$ to $T$:

$$\sum_{t=1}^{T}[F_t(v_t) - F_{t+1}(v_{t+1}) + \tilde{\ell}_t(v_t)] \leq \sum_{t=1}^{T}\frac{\|g_t\|_\infty^2}{2\lambda_t} + \sum_{t=1}^{T}(\psi_t(v_t) - \psi_{t+1}(v_{t+1})).$$

The second sum on the right-hand side is a telescoping sum:

$$\sum_{t=1}^{T}(\psi_t(v_t) - \psi_{t+1}(v_{t+1})) = (\psi_1(v_1) - \psi_2(v_2)) + \cdots + (\psi_T(v_T) - \psi_{T+1}(v_{T+1})) = \psi_1(v_1) - \psi_{T+1}(v_{T+1}).$$

Since $\psi_1(v_1) = 0$, this sum equals $-\psi_{T+1}(v_{T+1})$. Substituting this back into the sum bound:

$$\sum_{t=1}^{T}[F_t(v_t) - F_{t+1}(v_{t+1}) + \tilde{\ell}_t(v_t)] \leq \sum_{t=1}^{T}\frac{\|g_t\|_\infty^2}{2\lambda_t} - \psi_{T+1}(v_{T+1}).$$

Now, substitute this bound into the FTRL guarantee:

$$\sum_{t=1}^{T}(\tilde{\ell}_t(v_t) - \tilde{\ell}_t(u)) \leq \psi_{T+1}(u) + \sum_{t=1}^{T}\frac{\|g_t\|_\infty^2}{2\lambda_t} - \psi_{T+1}(v_{T+1}) + (F_{T+1}(v_{T+1}) - F_{T+1}(u)).$$

Since $v_{T+1} = \arg\min_{v \in V} F_{T+1}(v)$, we have $F_{T+1}(v_{T+1}) \leq F_{T+1}(u)$, so $F_{T+1}(v_{T+1}) - F_{T+1}(u) \leq 0$.

$$\sum_{t=1}^{T}(\tilde{\ell}_t(v_t) - \tilde{\ell}_t(u)) \leq \psi_{T+1}(u) - \psi_{T+1}(v_{T+1}) + \sum_{t=1}^{T}\frac{\|g_t\|_\infty^2}{2\lambda_t}.$$

Now we bound the terms on the right-hand side. First, consider the difference of the regularizer terms $\psi_{T+1}(u) - \psi_{T+1}(v_{T+1})$. For any $v \in V \subseteq [0,1]^d$, $\psi_{T+1}(v) = \frac{1}{\eta}\sum_{s=1}^{T} v^{(s)} \log(v^{(s)} + 1)$. Since $v^{(s)} \in [0,1]$, $0 \leq v^{(s)} \log(v^{(s)} + 1) \leq \log 2$. Thus, $0 \leq \psi_{T+1}(v) \leq \frac{T \log 2}{\eta}$ for any $v \in V$. Therefore, $\psi_{T+1}(u) - \psi_{T+1}(v_{T+1}) \leq \psi_{T+1}(u) \leq \frac{T \log 2}{\eta}$.

Next, consider the sum of gradient terms $\sum_{t=1}^{T}\frac{\|g_t\|_\infty^2}{2\lambda_t}$. From Lemma A.7, $\lambda_t = \frac{3}{4\eta(t-1)}$ for $t > 1$. The subgradient $g_t \in \partial\tilde{\ell}_t(v_t)$. Since $\tilde{\ell}_t(v) = \frac{1}{t-1}\sum_{s=1}^{t-1} \ell(v^{(s)}, r_t(x_s))$ and $\ell$ is L-Lipschitz, the infinity norm of $g_t$ is bounded by $\|g_t\|_\infty \leq \frac{L}{t-1}$ for $t > 1$. Substituting these bounds:

$$\sum_{t=2}^{T}\frac{\|g_t\|_\infty^2}{2\lambda_t} \leq \sum_{t=2}^{T}\frac{(L/(t-1))^2}{2 \cdot \frac{3}{4\eta(t-1)}} = \sum_{t=2}^{T}\frac{L^2}{(t-1)^2} \cdot \frac{2\eta(t-1)}{3} = \sum_{t=2}^{T}\frac{2\eta L^2}{3(t-1)}.$$

$$\sum_{t=2}^{T}\frac{2\eta L^2}{3(t-1)} = \frac{2\eta L^2}{3}\sum_{t=2}^{T}\frac{1}{t-1} = \frac{2\eta L^2}{3}\sum_{k=1}^{T-1}\frac{1}{k}.$$

Using the bound $\sum_{k=1}^{T-1} \frac{1}{k} \le 1 + \log(T-1)$ for $T > 1$:

$$\sum_{t=1}^{T} \frac{\|g_t\|_{\star}^2}{2\lambda_t} \le \frac{2\eta L^2}{3}(1 + \log(T-1)).$$

Combining the bounds for the two terms:

$$\sum_{t=1}^{T} (\tilde{\ell}_t(v_t) - \tilde{\ell}_t(u)) \le \frac{T \log 2}{\eta} + \frac{2\eta L^2}{3}(1 + \log(T-1)).$$

If we choose $\eta = \sqrt{\frac{T}{L^2 \log T}}$, the empirical regret is bounded by:

$$\sum_{t=1}^{T} (\tilde{\ell}_t(v_t) - \tilde{\ell}_t(u)) \le \frac{T \log 2}{\sqrt{\frac{T}{L^2 \log T}}} + \frac{2\sqrt{\frac{T}{L^2 \log T}} L^2}{3}(1 + \log(T-1)) = O(L\sqrt{T \log T}).$$

Thus, with this choice of $\eta$, the empirical regret is bounded by $O(L\sqrt{T \log T})$.

$\square$

**Corollary A.8** (Corollary 7.7 of Orabona (2023)). *Let $f : \mathbb{R}^d \to (-\infty, +\infty]$ be closed, proper, subdifferentiable, and $\mu$-strongly convex with respect to a norm $\|\cdot\|$ over its domain. Let $v^{\star} = \arg\min_v f(v)$. Then, for all $v \in \text{dom } \partial f$, and $g \in \partial f(v)$, we have*

$$f(v) - f(v^{\star}) \le \frac{1}{2\mu} \|g\|_{\star}^2.$$

**Theorem A.9** (Theorem 6.12 of Orabona (2023)). *Let $f : \mathbb{R}^d \to (-\infty, +\infty]$ be proper. Then $v^{\star} \in \arg\min_{v \in \mathbb{R}^d} f(v)$ iff $0 \in \partial f(v^{\star})$.*

*Proof of Lemma A.6.* Let $V^{(t-1)} = \{v^{(1:t-1)} : v \in V\} \subseteq [0,1]^{t-1}$. Define $\bar{F}_t : V^{(t-1)} \to \mathbb{R}$ such that $\bar{F}_t(v^{(1:t-1)}) = F_t(v)$ for $v \in V$. Note that $F_t(v) = \psi_t(v) + \sum_{s=1}^{t-1} \tilde{\ell}_s(v)$. Since $\psi_t$ depends only on $v^{(1:t-1)}$ and $\tilde{\ell}_s$ (for $s < t$) depends on $v^{(1:s-1)} \subseteq v^{(1:t-1)}$, $F_t$ indeed depends only on $v^{(1:t-1)}$. Similarly, we define $\bar{\ell}_t : V^{(t-1)} \to \mathbb{R}$ such that $\bar{\ell}_t(v^{(1:t-1)}) = \tilde{\ell}_t(v)$ for $v \in V, t \in [T]$. By Lemma A.7, $\psi_t$ is closed, subdifferentiable, and strongly convex with parameter $\lambda_t = \frac{3}{4\eta(t-1)}$ with respect to the $\ell_1$ norm on $V^{(t-1)}$ (for $t > 1$). As a sum of a strongly convex function ($\psi_t$) and convex functions ($\tilde{\ell}_s$), $\bar{F}_t$ is also strongly convex with parameter $\lambda_t = \frac{3}{4\eta(t-1)}$ on $V^{(t-1)}$ (for $t > 1$). The function $(v^{(1:t-1)}) \mapsto F_t(v) + \tilde{\ell}_t(v)$ is closed, subdifferentiable, and strongly convex with parameter $\lambda_t$ with respect to the $\ell_1$ norm on $V^{(t-1)}$.

$$
\begin{aligned}
&F_t(v_t) - F_{t+1}(v_{t+1}) + \tilde{\ell}_t(v_t) \\
&= (F_t(v_t) + \tilde{\ell}_t(v_t)) - (F_t(v_{t+1}) + \tilde{\ell}_t(v_{t+1})) + \psi_t(v_{t+1}) - \psi_{t+1}(v_{t+1}) \quad \text{(Rearranging terms)} \\
&= (\bar{F}_t(v_t^{(1:t-1)}) + \bar{\ell}_t(v_t^{(1:t-1)})) - (\bar{F}_t(v_{t+1}^{(1:t-1)}) + \bar{\ell}_t(v_{t+1}^{(1:t-1)})) + \psi_t(v_{t+1}) - \psi_{t+1}(v_{t+1})
\end{aligned}
$$

Recall that $\bar{F}_t$ is also strongly convex with parameter $\lambda_t = \frac{3}{4\eta(t-1)}$ on $V^{(t-1)}$ (for $t > 1$), thus, if we define $v_{t+1}^{*,(1:t-1)} := \arg\min_{v \in V^{(1:t-1)}} \{\bar{F}_t(v) + \bar{\ell}_t(v)\}$. By Corollary A.8, $(\bar{F}_t(v_t^{(1:t-1)}) + \bar{\ell}_t(v_t^{(1:t-1)})) - (\bar{F}_t(v_{t+1}^{*,(1:t-1)}) + \bar{\ell}_t(v_{t+1}^{*,(1:t-1)})) \le \frac{\|g_t\|_{\infty}^2}{2\lambda_t}$ where $g_t \in \delta(\bar{F}_t + \bar{\ell}_t)(v_t^{(1:t-1)})$. Now, use the fact that $v_t^{(1:t-1)} \in \arg\min_{v \in V^{(t-1)}} F_t(v)$, which by Theorem A.9 implies $0 \in \delta\bar{F}_t(v_t^{(1:t-1)})$, which implies $g_t \in \delta\bar{\ell}_t(v_t^{(1:t-1)})$. And because $\tilde{\ell}_t$ only depends on the first $t-1$ coordinates, then $\delta\bar{\ell}_t = \delta\tilde{\ell}_t$. Thus, for any $g_t \in \partial\tilde{\ell}_t(v_t)$, we have

$$F_t(v_t) - F_{t+1}(v_{t+1}) + \tilde{\ell}_t(v_t) \le \frac{\|g_t\|_{\infty}^2}{2\lambda_t} + \psi_t(v_{t+1}) - \psi_{t+1}(v_{t+1}).$$

for $\lambda_t = \frac{3}{4\eta(t-1)}$

$\square$

*Proof of Lemma A.7.* Let $v \in V^{(t-1)} \subseteq [0,1]^{t-1}$, so $v$ is a vector $(v^{(1)}, \dots, v^{(t-1)})$ with $v^{(s)} \in [0,1]$ for $s = 1, \dots, t-1$. Let $u = y - x$ where $x, y \in V^{(t-1)}$, so $u$ is a vector $(u_1, \dots, u_{t-1}) \in \mathbb{R}^{t-1}$. The

function is $\psi_t(v) = \frac{1}{\eta} \sum_{s=1}^{t-1} v^{(s)} \log(v^{(s)} + 1)$. Let $f(w) = w \log(w + 1)$. The second derivative is $f''(w) = \frac{1}{w+1} + \frac{1}{(w+1)^2}$. For $w \in [0, 1]$, $w + 1 \in [1, 2]$, so $f''(w) \geq \frac{1}{2} + \frac{1}{4} = \frac{3}{4}$.

The Hessian matrix $\nabla^2 \psi_t(v)$ is a $(t - 1) \times (t - 1)$ diagonal matrix with entries $(\nabla^2 \psi_t(v))_{ss} = \frac{1}{\eta} f''(v^{(s)})$ for $s = 1, \ldots, t - 1$.

To show $\lambda_t$-strong convexity with respect to the $\ell_1$ norm, we show $u^T \nabla^2 \psi_t(v) u \geq \lambda_t \|u\|_1^2$ for all $v \in V^{(t-1)}$ and $u \in \mathbb{R}^{t-1}$.

$$u^T \nabla^2 \psi_t(v) u = \sum_{s=1}^{t-1} (\nabla^2 \psi_t(v))_{ss} u_s^2 = \sum_{s=1}^{t-1} \frac{1}{\eta} f''(v^{(s)}) u_s^2$$

Since $v^{(s)} \in [0, 1]$ for $s = 1, \ldots, t - 1$, $f''(v^{(s)}) \geq \frac{3}{4}$.

$$u^T \nabla^2 \psi_t(v) u \geq \sum_{s=1}^{t-1} \frac{1}{\eta} \left(\frac{3}{4}\right) u_s^2 = \frac{3}{4\eta} \sum_{s=1}^{t-1} u_s^2 = \frac{3}{4\eta} \|u\|_2^2$$

We use the relationship between the $\ell_2$ and $\ell_1$ norms in $\mathbb{R}^{t-1}$: $\|u\|_2^2 \geq \frac{1}{t-1} \|u\|_1^2$. This holds for $t - 1 > 0$, i.e., $t > 1$.

$$\frac{3}{4\eta} \|u\|_2^2 \geq \frac{3}{4\eta} \left(\frac{1}{t-1} \|u\|_1^2\right) = \frac{3}{4\eta(t-1)} \|u\|_1^2$$

Thus, $u^T \nabla^2 \psi_t(v) u \geq \frac{3}{4\eta(t-1)} \|u\|_1^2$. Comparing this with the strong convexity condition $u^T \nabla^2 \psi_t(v) u \geq \lambda_t \|u\|_1^2$, we can choose $\lambda_t = \frac{3}{4\eta(t-1)}$.

$\square$

**Lemma A.10** (Summing Rademacher complexities over prefixes). Let $\mathcal{F} \subseteq [0, 1]^X$ be a class of functions and let $\mathrm{rad}_m(\mathcal{F})$ denote its empirical Rademacher complexity on $m$ samples. Then

$$\sum_{t=2}^{T} \mathrm{rad}_{t-1}(\mathcal{F}) \leq \tilde{O}(T \cdot \mathrm{rad}_T(\mathcal{F})),$$

where $\tilde{O}(\cdot)$ hides universal constants and logarithmic factors in $T$.

*Proof.* We use the near-tight characterization of Rademacher complexity via Dudley's entropy integral. Results of Sridharan (2010) imply that for any $m$,

$$\mathrm{rad}_m(\mathcal{F}) \leq \tilde{O}\left(\inf_{\alpha>0} \left\{\alpha + \frac{1}{\sqrt{m}} \int_\alpha^1 \sqrt{\mathrm{fat}_\delta(\mathcal{F}) \log \frac{2}{\delta}} \, d\delta\right\}\right),$$

where $\mathrm{fat}_\delta(\mathcal{F})$ denotes the fat-shattering dimension of $\mathcal{F}$ at scale $\delta$.

Applying this bound with $m = t - 1$ and summing from $t = 2$ to $T$ gives

$$\sum_{t=2}^{T} \mathrm{rad}_{t-1}(\mathcal{F}) \leq \tilde{O}\left(\sum_{t=2}^{T} \inf_{\alpha>0} \left\{\alpha + \frac{1}{\sqrt{t-1}} \int_\alpha^1 \sqrt{\mathrm{fat}_\delta(\mathcal{F}) \log \frac{2}{\delta}} \, d\delta\right\}\right).$$

Pulling the infimum outside and summing terms yields

$$\leq \tilde{O}\left(\inf_{\alpha>0} \left\{(T - 1)\alpha + \left(\sum_{t=2}^{T} \frac{1}{\sqrt{t-1}}\right) \int_\alpha^1 \sqrt{\mathrm{fat}_\delta(\mathcal{F}) \log \frac{2}{\delta}} \, d\delta\right\}\right).$$

Since $\sum_{s=1}^{T-1} s^{-1/2} \leq 2\sqrt{T}$, this is at most

$$\tilde{O}\left(T \cdot \inf_{\alpha>0} \left\{\alpha + \frac{1}{\sqrt{T}} \int_\alpha^1 \sqrt{\mathrm{fat}_\delta(\mathcal{F}) \log \frac{2}{\delta}} \, d\delta\right\}\right).$$

Applying the same Dudley-type bound again with $m = T$ shows that the term in braces is $\tilde{O}(\mathrm{rad}_T(\mathcal{F}))$, yielding

$$\sum_{t=2}^{T} \mathrm{rad}_{t-1}(\mathcal{F}) \leq \tilde{O}(T \cdot \mathrm{rad}_T(\mathcal{F})).$$

$\square$

A.2   DEFERRED PROOFS FROM SECTION 1.2

To aid with their introduction and analysis of the distribution-dependent sequential Rademacher complexity, Rakhlin et al. (2011) introduce the following notation:

They define the *selector function* $\chi : \mathcal{X} \times \mathcal{X} \times \{\pm 1\} \to \mathcal{X}$, which selects one of two elements based on a binary sign $\epsilon$:

$$\chi(x, x', \epsilon) = \begin{cases} x' & \text{if } \epsilon = 1 \\ x & \text{if } \epsilon = -1 \end{cases}$$

In the context of sequences where $x_t$ and $x'_t$ implicitly depend on previous $\epsilon$ values, the shorthand $\chi_t(\boldsymbol{\epsilon}) := \chi(x_t(\epsilon_{1:t-1}), x'_t(\epsilon_{1:t-1}), \epsilon_t)$. This notation indicates that $\chi_t$ chooses either $x_t$ or $x'_t$ at time step $t$, depending on the value of $\epsilon_t$ within the path $\boldsymbol{\epsilon} = (\epsilon_1, \dots, \epsilon_T)$. The terms $x_t(\epsilon_{1:t-1})$ and $x'_t(\epsilon_{1:t-1})$ represent elements at depth $t$ along a specific path determined by the preceding $\epsilon$ values.

A *$Z$-valued tree of depth $T$* is a sequence of $T$ mappings, $(\mathbf{z}_1, \dots, \mathbf{z}_T)$. Each mapping $\mathbf{z}_t : \{\pm 1\}^{t-1} \to Z$ assigns a value from set $Z$ to a specific node at depth $t$. The node's position is uniquely determined by a sequence of prior choices, $(\epsilon_1, \dots, \epsilon_{t-1}) \in \{\pm 1\}^{t-1}$. A complete sequence $\boldsymbol{\epsilon} = (\epsilon_1, \dots, \epsilon_T) \in \{\pm 1\}^T$ defines a unique path from the root to a leaf of the tree. For conciseness, $\mathbf{z}_t(\epsilon_{1:t-1})$ is shorthand for $\mathbf{z}_t(\epsilon_1, \dots, \epsilon_{t-1})$.

Given an underlying joint distribution $\mathbf{p}$ (over $T$ length sequences of observations from $\mathcal{X}$), we define a *probability tree* $\rho = (\rho_1, \dots, \rho_T)$. This tree generates sequences of pairs of elements $(\mathbf{x}, \mathbf{x}') = ((x_1, x'_1), \dots, (x_T, x'_T))$. Each $\rho_t(\epsilon_{1:t-1})$ is a conditional probability distribution that determines $(x_t, x'_t)$ given the preceding pairs $(x_1, x'_1), \dots, (x_{t-1}, x'_{t-1})$. The crucial aspect is how this conditioning is performed:

$$\rho_t(\epsilon_{1:t-1})((x_t, x'_t)|(x_{1:t-1}, x'_{1:t-1})) = \mathbf{p}_t((\chi_s(\epsilon_s))_{s=1}^{t-1})((x_t, x'_t)|(x_{1:t-1}, x'_{1:t-1})) \tag{4}$$

Here, $\mathbf{p}_t((\chi_s(\epsilon_s))_{s=1}^{t-1})$ denotes the conditional distribution for $(x_t, x'_t)$ derived from $\mathbf{p}$, given the history sequence formed by dynamically applying the selector function at each step: $(\chi_1(\epsilon_1), \dots, \chi_{t-1}(\epsilon_{t-1}))$. This means the generation of each pair $(x_t, x'_t)$ depends on a history that dynamically selects between $x_s$ and $x'_s$ based on the Rademacher variables $\epsilon_s$.

**Definition A.1** (Definition 2 of Rakhlin et al. (2011))**.** The distribution-dependent sequential Rademacher complexity of a function class $\mathcal{F} \subseteq \mathbb{R}^{\mathcal{X}}$ is defined as

$$\Re_T(\mathcal{F}, \mathbf{p}) \triangleq \mathbb{E}_{(\mathbf{x}, \mathbf{x}') \sim \rho} \mathbb{E}_{\boldsymbol{\epsilon}} \left[ \sup_{f \in \mathcal{F}} \sum_{t=1}^{T} \epsilon_t f(\chi_t(\boldsymbol{\epsilon})) \right]$$

where $\boldsymbol{\epsilon} = (\epsilon_1, \dots, \epsilon_T)$ is a sequence of i.i.d. Rademacher random variables and $\rho$ is the probability tree associated with $\mathbf{p}$ as explained in Equation (4).

**Lemma A.11** (Lemma 17 of Rakhlin et al. (2011))**.** Fix a class $\mathcal{F} \subseteq \mathbb{R}^{\mathcal{X}}$ and a function $\phi : \mathbb{R} \times \mathcal{Y} \to \mathbb{R}$. Given a distribution $p$ over $\mathcal{X}$, let $\mathfrak{P}$ consist of all joint distributions $\mathbf{p}$ such that the conditional distribution $p_t^{x,y}(x_t, y_t|x^{t-1}, y^{t-1})$ can be written as $p(x_t) \times p_t(y_t|x^{t-1}, y^{t-1}, x_t)$ for some conditional distribution $p_t$. Then,

$$\sup_{\mathbf{p} \in \mathfrak{P}} \Re_T(\phi(\mathcal{F}), \mathbf{p}) \le \mathbb{E}_{\mathbf{x}_1, \dots, \mathbf{x}_T \sim p, \mathbf{y} \sim \mathbf{p}} \left[ \mathbb{E}_{\boldsymbol{\epsilon}} \sup_{f \in \mathcal{F}} \sum_{t=1}^{T} \epsilon_t \phi(f(x_t), y_t(\boldsymbol{\epsilon})) \right].$$

**Lemma A.12** (Lemma 18 of Rakhlin et al. (2011))**.** Fix a class $\mathcal{F} \subseteq [-1, 1]^{\mathcal{X}}$ and a function $\phi : [-1, 1] \times \mathcal{Y} \to \mathbb{R}$. Assume, for all $y \in \mathcal{Y}$, $\phi(\cdot, y)$ is a Lipschitz function with a constant $L$. Let $\mathfrak{P}$ be as in Lemma A.11. Then, for any $\mathbf{p} \in \mathfrak{P}$,

$$\Re_T(\phi(\mathcal{F}), \mathbf{p}) \le L \Re_T(\mathcal{F}, p).$$

**Proposition 1.3.** Let $\mathcal{H}$ be a class of hypothesis functions and $\ell$ be a loss function that is $L$-Lipschitz in the first parameter. Let $x_1, x_2, \dots, x_T$ be a sequence of i.i.d samples from a fixed distribution $\mathcal{D}$. Let $r_1, r_2, \dots, r_T \in [0, 1]^{\mathcal{X}}$ be a sequence of functions where $r_t$ depends only on $x_1, \dots, x_{t-1}$ (and potentially prior adversarial choices). The following holds with probability at least $1 - \delta$ over the draw of $x_1, \dots, x_T$, for all $h \in \mathcal{H}$:

$$\left| \frac{1}{T} \sum_{t=1}^{T} \ell(h(x_t), r_t(x_t)) - \frac{1}{T} \sum_{t=1}^{T} \mathbb{E}_{x \sim \mathcal{D}} [\ell(h(x), r_t(x))] \right| \le O\left( L \cdot \mathsf{rad}_T(\mathcal{H}) + L \sqrt{\frac{\log(T/\delta)}{T}} \right)$$

*Proof of Proposition 1.3.* Note that $\ell(h(x_t), r_t(x_t)) - \mathbb{E}_{\mathcal{D}}[\ell(h(x), r_t(x))]$ is a martingale difference sequence since $x_t$ is sampled after the choice of $r_t$ is made. We will apply the classic symmetrization technique, borrowing ideas from the proof of Theorem 3 of Rakhlin et al. (2011). We will consider a tangent sequence $\{x'\}_{t=1}^T$ that is drawn i.i.d from the distribution $\mathcal{D}$. Note that this tangent sequence is independent of $\{x\}_{t=1}^T$. For any sequence of $r_1, \ldots, r_T$, The LHS of the equation reduces to the following:

$$\mathbb{E}\left[\sup_{h \in \mathcal{H}} \left\{\frac{1}{T} \sum_{t=1}^T \ell(h(x_t), r_t(x_t)) - \frac{1}{T} \sum_{t=1}^T \ell(h(x'_t), r_t(x'_t))\right\}\right] \tag{5}$$

$$= \mathbb{E}_{(x_1,x'_1)\sim\mathcal{D}} \mathbb{E}_{(x_2,x'_2)\sim\mathcal{D}} \ldots \mathbb{E}_{(x_T,x'_T)\sim\mathcal{D}}\left[\sup_{h \in \mathcal{H}} \left\{\frac{1}{T} \sum_{t=1}^T \ell(h(x_t), r_t(x_t)) - \frac{1}{T} \sum_{t=1}^T \ell(h(x'_t), r_t(x'_t))\right\}\right] \tag{6}$$

$$\leq \sup_{r_1 \in \mathcal{R}} \mathbb{E}_{(x_1,x'_1)\sim\mathcal{D}} \sup_{r_2 \in \mathcal{R}(\cdot|x_1)} \mathbb{E}_{(x_2,x'_2)\sim\mathcal{D}} \ldots \tag{7}$$

$$\ldots \sup_{r_T \in \mathcal{R}(\cdot|x_1,\ldots,x_{T-1})} \mathbb{E}_{(x_T,x'_T)\sim\mathcal{D}}\left[\sup_{h \in \mathcal{H}} \left\{\frac{1}{T} \sum_{t=1}^T \ell(h(x_t), r_t(x_t)) - \frac{1}{T} \sum_{t=1}^T \ell(h(x'_t), r_t(x'_t))\right\}\right] \tag{8}$$

Now fix $\epsilon \in \{\pm 1\}^T$ and let $-\epsilon_t$ denote whether we switch $x_t$ with $x'_t$. Since these are from the same distribution, this does not affect the expectation over $\mathcal{D}$. Thus, the last equation simplifies to

$$\sup_{r_1 \in \mathcal{R}} \mathbb{E}_{(x_1,x'_1)\sim\mathcal{D}} \sup_{r_2 \in \mathcal{R}(\cdot|x_1(\epsilon_1))} \mathbb{E}_{(x_2,x'_2)\sim\mathcal{D}} \ldots \sup_{r_T \in \mathcal{R}(\cdot|x_1(\epsilon_1),\ldots,x_{T-1}(\epsilon_{T-1}))} \mathbb{E}_{(x_T,x'_T)\sim\mathcal{D}} \tag{9}$$

$$\left[\sup_{h \in \mathcal{H}} \left\{\frac{1}{T} \sum_{t=1}^T \epsilon_t \left(\ell(h(x_t), r_t(x_t)) - \ell(h(x'_t), r_t(x'_t))\right)\right\}\right] \tag{10}$$

Taking expectation over $\epsilon \in \{\pm 1\}^T$, we have that

$$\mathbb{E}\left[\sup_{h \in \mathcal{H}} \left\{\frac{1}{T} \sum_{t=1}^T \ell(h(x_t), r_t(x_t)) - \frac{1}{T} \sum_{t=1}^T \ell(h(x'_t), r_t(x'_t))\right\}\right] \tag{11}$$

$$\leq \sup_{r_1 \in \mathcal{R}} \mathbb{E}_{(x_1,x'_1)} \mathbb{E}_{\epsilon_1} \sup_{r_2 \in \mathcal{R}(\cdot|x_1(\epsilon_1))} \mathbb{E}_{(x_2,x'_2)} \mathbb{E}_{\epsilon_2} \ldots \sup_{r_T \in \mathcal{R}(\cdot|x_1(\epsilon_1),\ldots,x_{T-1}(\epsilon_{T-1}))} \mathbb{E}_{(x_T,x'_T)} \mathbb{E}_{\epsilon_T} \tag{12}$$

$$\left[\sup_{h \in \mathcal{H}} \left\{\frac{1}{T} \sum_{t=1}^T \epsilon_t \left(\ell(h(x_t), r_t(x_t)) - \ell(h(x'_t), r_t(x'_t))\right)\right\}\right] \tag{13}$$

The process above can be thought of as taking a path in a binary tree whose nodes are represented by functions $r \in \mathcal{R}$. At each step t, $r_t$ is chosen and then a coin is flipped and this determines whether $x_t$ or $x'_t$ is to be used in the following steps. We write the last expression concisely as

$$\sup_{\mathbf{r}} \mathbb{E}_{(x,x')\sim\mathcal{D}}\left[\sup_{h \in \mathcal{H}} \left\{\frac{1}{T} \sum_{t=1}^T \epsilon_t \left(\ell(h(x_t), r_t(x_t)) - \ell(h(x'_t), r_t(x'_t))\right)\right\}\right]$$

And this can be upper bounded by two times the distribution-dependent Rademacher complexity notion defined in Definition A.1

$$\sup_{\mathbf{r}} \mathbb{E}_{(x,x')\sim\mathcal{D}}\left[\sup_{h \in \mathcal{H}} \left\{\frac{1}{T} \sum_{t=1}^T \epsilon_t \left(\ell(h(x_t), r_t(x_t)) - \ell(h(x'_t), r_t(x'_t))\right)\right\}\right] \tag{14}$$

$$\leq 2 \sup_{\mathbf{r}} \mathbb{E}_{(x,x')\sim\mathcal{D}}\left[\sup_{h \in \mathcal{H}} \left\{\frac{1}{T} \sum_{t=1}^T \epsilon_t \ell(h(x_t), r_t(x_t))\right\}\right] \tag{15}$$

$$\leq 2 \sup_{\mathbf{p} \in \mathfrak{P}} \mathfrak{R}_T(\ell \circ \mathcal{H}, \mathbf{p}) \tag{16}$$

where $\mathfrak{P}$ consists of all joint distributions $\mathbf{p}$ such that the conditional distribution $p_t^{x,y}(x_t, y_t|x^{t-1}, y^{t-1})$ can be written as $p(x_t) \times p_t(y_t|x^{t-1}, y^{t-1}, x_t)$ for some conditional distribution $p_t$. Applying Lemma A.11 together with Lemma A.12 gives the desired result. To obtain the high probability version of the statement, we follow the same steps here replacing the expected Rademacher with high probability Rademacher as done in Lemma 4 of Rakhlin et al. (2015). □

# B   Deferred Proof from Section 4

*Proof of Lemma 3.1.* Let $p = |S|$. We order the elements of $S$ as $(s_1, \ldots, s_p)$. The feasible set is $\mathcal{K}_S = \{(h(s_1), \ldots, h(s_p)) \mid h \in \text{conv}(\mathcal{H})\}$. Since $h : [0, 1]^X$, $\mathcal{K}_S \subseteq [0, 1]^p$. The objective function is $G : \mathcal{K}_S \to \mathbb{R}$ defined as

$$G(z) = \eta \sum_{i=1}^{m} w_i \ell(z_{x_i}, y_i) \; + \; \sum_{s \in S} z_s \log(z_s + 1),$$

where $z \in \mathcal{K}_S$ and $z_s$ denotes the component of $z$ corresponding to $s \in S$. The function $G$ is well-defined and differentiable on $\mathcal{K}_S$. Its partial derivative with respect to $z_s$ for $s \in S$ is:

$$\frac{\partial G(z)}{\partial z_s} = \eta \sum_{i:x_i=s} w_i \frac{\partial \ell(z_s, y_i)}{\partial z_s} + \log(z_s + 1) + \frac{z_s}{z_s + 1}.$$

The Hessian of $G(z)$ is a diagonal matrix. The diagonal entry corresponding to $z_s$ is $\frac{\partial^2 G(z)}{\partial z_s^2}$.

$$\frac{\partial^2 G(z)}{\partial z_s^2} = \eta \sum_{i:x_i=s} w_i \frac{\partial^2 \ell(z_s, y_i)}{\partial z_s^2} + \frac{1}{z_s + 1} + \frac{1}{(z_s + 1)^2}.$$

Since $\ell$ is $\beta$-smooth, $|\frac{\partial^2 \ell}{\partial u^2}| \leq \beta$. For $z_s \in [0, 1]$, we have $\frac{1}{z_s+1} + \frac{1}{(z_s+1)^2} \leq 1 + 1 = 2$. Let $I(s) = \{i \in \{1, \ldots, m\} \mid x_i = s\}$. Then

$$\left| \frac{\partial^2 G(z)}{\partial z_s^2} \right| \leq \eta \sum_{i \in I(s)} |w_i| \left| \frac{\partial^2 \ell(z_s, y_i)}{\partial z_s^2} \right| + \left| \frac{1}{z_s + 1} + \frac{1}{(z_s + 1)^2} \right| \leq \eta \left( \sum_{i \in I(s)} |w_i| \right) \beta + 2.$$

Let $W_{\max} = \max_{s \in S} \sum_{i:x_i=s} |w_i|$. The maximum absolute value of the diagonal entries of the Hessian of $G(z)$ is bounded by $\eta W_{\max} \beta + 2$. Therefore, $G$ is $\beta_G$-smooth with $\beta_G = \eta W_{\max} \beta + 2$.

The set $\mathcal{K}_S \subseteq [0, 1]^p$. The $\ell_2$-diameter of $\mathcal{K}_S$ is at most the $\ell_2$-diameter of the hypercube $[0, 1]^p$, which is $\sqrt{\sum_{j=1}^{p} (1 - 0)^2} = \sqrt{p} = \sqrt{|S|}$. Let $R = \sqrt{|S|}$.

Applying Lemma 3.2 to $G$ on $\mathcal{K}_S$:

$$G(z_T) - G(z^*) \leq \frac{2\beta_G R^2}{T + 1} \leq \frac{2(\eta W_{\max} \beta + 2)|S|}{T + 1}.$$

To achieve $G(z_T) - G(z^*) < \epsilon$, we need

$$\frac{2(\eta W_{\max} \beta + 2)|S|}{T + 1} \leq \epsilon,$$

which implies

$$T + 1 \geq \frac{2|S|(\eta W_{\max} \beta + 2)}{\epsilon}.$$

Thus, $T = O\left( \frac{|S|(\eta W_{\max} \beta + 1)}{\epsilon} \right)$. $\square$

# C   Deferred Proofs from Section 4

*Proof of Corollary 4.1.* Let $S = \{x_1, \ldots, x_m\}$ be a set of $m$ i.i.d. samples drawn from $\mathcal{D}$. We will use the hybrid learner algorithm (Algorithm 1) with $T = m$ steps. The samples for the hybrid learner are the drawn samples $x_1, \ldots, x_m$. At each step $t \in \{1, \ldots, m\}$, the hybrid learner outputs a hypothesis $h_t \in \text{conv}(\mathcal{H})$. We define the adversary function for step $t$ of the hybrid learner as the empirical best response to $h_t$ on the full sample set $S$:

$$r_t = \text{argmax}_{r \in \mathcal{R}} \frac{1}{t - 1} \sum_{i=1}^{t-1} u(h_t(x_i), r(x_i)).$$

This sequence of adversaries $r_1, \ldots, r_m$ is provided to the hybrid learner. $r_t$ is chosen only using the first $t - 1$ samples observed by the algorithm in order to preserve any martingale properties of

the algorithm [2]. At timestep $t$, after outputting $h_t$ and observing $x_t$, the hybrid learner receives $r_t$ (computed as the empirical best response to $h_t$ on $S_{t-1}$) as the adversary function for the current step.

Let $h^* \in \text{conv}(\mathcal{H})$ be the optimal hypothesis in expectation against the sequence $r_1, \ldots, r_m$: $h^* = \arg\min_{h \in \text{conv}(\mathcal{H})} \sum_{t=1}^m \mathbb{E}[u(h(x), r_t(x))]$. The hybrid learner theorem (Theorem 2.1) guarantees that with probability at least $1 - \delta'$,

$$\sum_{t=1}^m \mathbb{E}[u(h_t(x), r_t(x))] \leq \min_{h \in \text{conv}(\mathcal{H})} \sum_{t=1}^m \mathbb{E}[u(h(x), r_t(x))] + 2m \cdot \text{rad}_m(\mathcal{F}) + O\left(L\sqrt{m \log m}\right).$$

Consider the average policies $h_A = \bar{h} = \frac{1}{m} \sum_{t=1}^m h_t$ and $r_A = \bar{r} = \frac{1}{m} \sum_{t=1}^m r_t$. By convexity of $u$ in the first argument, $\mathbb{E}[u(h_A, r)] \leq \frac{1}{m} \sum_{t=1}^m \mathbb{E}[u(h_t, r)]$. By concavity of $u$ in the second argument, $\mathbb{E}[u(h, r_A)] \geq \frac{1}{m} \sum_{t=1}^m \mathbb{E}[u(h, r_t)]$.

Consider the saddle point gap for $(h_A, r_A)$: $\max_{r \in \text{conv}(\mathcal{R})} \mathbb{E}[u(h_A, r)] - \min_{h \in \text{conv}(\mathcal{H})} \mathbb{E}[u(h, r_A)]$.

$$\max_{r \in \text{conv}(\mathcal{R})} \mathbb{E}[u(h_A, r)] \leq \frac{1}{m} \sum_{t=1}^m \max_{r \in \text{conv}(\mathcal{R})} \mathbb{E}[u(h_t, r)].$$

$$\min_{h \in \text{conv}(\mathcal{H})} \mathbb{E}[u(h, r_A)] \geq \min_{h \in \text{conv}(\mathcal{H})} \frac{1}{m} \sum_{t=1}^m \mathbb{E}[u(h, r_t)].$$

So,

$$\max_{r \in \text{conv}(\mathcal{R})} \mathbb{E}[u(h_A, r)] - \min_{h \in \text{conv}(\mathcal{H})} \mathbb{E}[u(h, r_A)] \leq \frac{1}{m} \sum_{t=1}^m \left( \max_{r \in \text{conv}(\mathcal{R})} \mathbb{E}[u(h_t, r)] - \min_{h \in \text{conv}(\mathcal{H})} \mathbb{E}[u(h, r_t)] \right).$$

Applying Lemma 2.3, we have that for all $t > 1$, for all $r \in \mathcal{R}$

$$\left| \mathbb{E}_{x \sim \mathcal{D}}[u(h_t(x), r(x))] - \frac{1}{t-1} \sum_{s=1}^{t-1} u(h_t(x_s), r(x_s)) \right| \leq 2\text{rad}_{t-1}(\mathcal{F})] + \sqrt{\frac{\log(2T/\delta)}{t-1}}$$

Thus,

$$\frac{1}{m} \sum_{t=1}^m \max_{r \in \text{conv}(\mathcal{R})} \mathbb{E}[u(h_t, r)] \leq \frac{1}{m} \sum_{t=1}^m \mathbb{E}[u(h_t, r_t)] - \sum_{t=1}^m 2\text{rad}_{t-1}(\mathcal{F})] - \sum_{t=1}^m \sqrt{\frac{\log(2m/\delta)}{t-1}}$$

Applying the regret guarantee from Theorem 2.1 to $\sum_{t=1}^m \mathbb{E}[u(h_t, r_t)] - \min_{h \in \text{conv}(\mathcal{H})} \mathbb{E}[u(h, r_t)]$ gives the desired result.

The total running time is $\text{poly}(m) \cdot$ (cost of $\mathcal{H}$ ERM oracle) $+ m \cdot$ (cost of $\mathcal{R}$ best response oracle). This is oracle-efficient in $\text{poly}(m)$. $\qquad\square$

---

[2] although Theorem 1 doesn't rely on the martingale nature of the data.