# OpenReview forum: "Oracle-efficient Hybrid Learning with Constrained Adversaries"
_ICLR.cc/2026/Conference — ICLR 2026 Poster_

### Official Review · Reviewer_SZPF · 2025-10-26

**Soundness:** 3
**Presentation:** 3
**Contribution:** 3
**Rating:** 6
**Confidence:** 2

**Summary:**

This paper studies hybrid online learning where features are drawn i.i.d. from an unknown distribution, while labels are chosen by an adversary. Different from the fully adversarial case, the adversary is constrained to pick labels from a fixed function class, and the learner competes with a hypothesis class under a Lipschitz abd convex loss function. The main result of this paper is to propose an oracle-efficient algorithm with a high-probability regret guarantee. The proposed algorithm is built upon FTRL framework with the entropy regularizer, implemented via a Frank–Wolfe reduction that calls a linear‑optimization oracle over hypothesis class.

**Strengths:**

- Prior work either attains optimal rates but is computationally intractable, or is oracle‑efficient but statistically suboptimal. This paper achieves near‑optimal rates and oracle efficiency by constraining the adversar, which could be an interesting middle ground.

- Authors propose a FTRL-based algorithm with entropy regularizer, and theoratically show that the proposed algorithm achieves a statistically near-optimal up to the dependence on the adversary’s constraint set $\mathcal{R}$.

- Proposition 1.3 addresses uniform convergence when the functions $r_t$ are chosen adaptively yet independently of $x_t$. This result could be a good contribution of independent interest.

- This paper is easy-to-follow and well-written. Though I am not quite familiar with the context, I appreciate detailed and clear explainations provided by authors.

**Weaknesses:**

- The tightness of the regret bound remains unclear to me, especially $T* rad_T(\ell \circ  \mathcal{H} \times \mathcal{R})$.

- The algorithm runs in $O(T^2)$ time per round and makes $O(T^2)$ calls to a linear optimization oracle, both of which seem to be pretty large.

- In line 265, $\log(a+1)$ is strongly concave on its domain rather than strongly convex. It should be that $a*log(a+1)$ is strongly convex.

**Questions:**

- I am not sure whether imposing the Lipschitz assumption on loss function is common or not. Can authors justify this assumption?

- The algorithm needs to have access of the loss function. I am curious what if the learner does not have the knowledge of the loss function, but only has bandit feedback (i.e., observe the loss of the chosen one).

---

> ### Author Response · Authors · 2025-12-04
>
> We thank the reviewer for the thoughtful review.
>
> On the tightness of the regret bound: Our bounds are tight up to dependence on the complexity of the adversary's class since classical statistical learning implies a lower bound of $T* rad_T(\ell \circ \mathcal{H})$. Removing this dependence on R remains a direction of future research.
>
> Thank you for identifying the mistake in line 265, we will fix that in the next version.
>
> On the dependence of the run time of the algorithm: Our key contribution is ensuring a poly(T) algorithm, a significant improvement from O(|H|) which you get from naively running multiplicative weight on a cover of the hypothesis class. Improving this polynomial dependence of T remains a direction for future work.
>
> On lipschitzness of the loss function: Yes this is a standard assumption in the hybrid and smoothed online learning literature.
>
> On access to loss function and bandit feedback: We think this is a great question for future study.

---

### Official Review · Reviewer_KxDd · 2025-10-29

**Soundness:** 3
**Presentation:** 3
**Contribution:** 3
**Rating:** 8
**Confidence:** 4

**Summary:**

This paper studies the problem of *hybrid online learning*, where the feature vectors are drawn i.i.d. from an unknown distribution, while the labels are chosen adversarially. It focuses on a structured setting in which the labels are generated by a function belonging to a restricted class. Without restrictions on the labeling functions, Wu et al. (2024) showed that an oracle-efficient online algorithm exists, though its regret bound exhibits a gap compared to the case where the data distribution is known. This paper shows that if the labeling functions are drawn from a class whose complexity is no bigger than that of the hypothesis class, one can achieve the minimax-optimal regret.

The main proof technique is based on constructing a *surrogate loss* using the previously observed samples and reducing the hybrid online learning problem to an instance of online convex optimization. The paper then introduces a *truncated entropy regularizer* and proves that applying FTRL with this regularizer yields a *pointwise* regret bound of $\tilde{O}(\sqrt{T})$. This pointwise guarantee is subsequently converted into a *high-probability* regret bound for the original hybrid learning problem via standard martingale concentration arguments. Finally, since vanilla FTRL requires convexity of the hypothesis class, the authors resolve this issue by employing a Frank–Wolfe–based oracle reduction.

**Strengths:**

1. I find the idea of reducing the hybrid online learning problem to an OCO problem quite interesting. This could inspire researchers to explore similar reductions for more complex hybrid settings, such as the smoothed adversary model.

2. As far as I understand, the truncated entropy regularizer is novel and may have broader application scenarios beyond this particular setting.

3. The authors demonstrated the usefulness of their constrained labeling-function formulation in the context of finding saddle points in minimax games, which I find convincing.

**Weaknesses:**

1. Although the paper provides a use case for the constrained labeling-function setting in games, it still feels somewhat restrictive, especially since the prior result by Wu et al. (2024) does not rely on such constraints. It would strengthen the paper if the authors could present additional examples where similar constraints arise naturally from structural properties of the problem.

2. The paper claims that the obtained regret bound is “near-optimal.” I am not entirely sure how this should be interpreted. In particular, if the VC dimension of $R$ is unbounded, the bound becomes vacuous, even though we know that sublinear regret is still achievable in that case.

**Questions:**

Can the OCO problem and its oracle-efficient solution be stated more generally? As far as I can see, the specific form of the loss does not matter, one only needs that each $\ell_t$ depends only on the first $t-1$ coordinates of $v$ and is convex and Lipschitz with respect to the $\ell_1$ norm.

---

> ### Author Response · Authors · 2025-12-04
>
> We thank the reviewer for their review. Yes indeed, the OCO problem and its oracle-efficient solution can be stated more generally. One only needs that each loss \ell_t depends on just the first t-1 coordinates of v and is convex and Lipschitz with respect to the l1-norm as you have correctly pointed out.

---

### Official Review · Reviewer_JWkH · 2025-11-01

**Soundness:** 2
**Presentation:** 2
**Contribution:** 3
**Rating:** 4
**Confidence:** 2

**Summary:**

The paper studies hybrid online learning where features are i.i.d. from an unknown distribution $D$ but labels are chosen adversarially from a constrained class $\mathcal R \subset [0,1]^X$. Given a linear optimization oracle over the hypothesis class $\mathcal{H} \subset [0,1]^X$, the authors design an ERM-oracle-efficient algorithm and prove a high-probability regret bound. The construction includes on (i) an approximate FTRL scheme over the “projection” of $\mathcal H$ onto seen samples, using a time-varying shifted entropy regularizer (to get strong convexity on the first $t-1$ coordinates), and (ii) a Frank–Wolfe reduction so the regularized ERM calls can be implemented with a linear optimization oracle. A uniform-convergence argument then upgrades in-expectation control to the realized-sample regret bound. The paper also gives an application to approximate equilibria in certain stochastic zero-sum games with separable structure (payoff $u(h(x),r(x))$).

**Strengths:**

- The considered problem is well-motivated and has real-world applications.
- The designed algorithms are intuitive and easy to implement. The algorithm is also computationally efficient compared to previous works.

**Weaknesses:**

- One concern is about the construction of FTRL. Specifically, I am not sure the role of the entropy regularizer is under-motivated. In the main text, the “truncated” entropy $v\mapsto \sum_s v(s)\log(v(s)+1)$ is chosen because $\log(1+a)$ is uniformly strongly convex on $[0,1]$, giving strong convexity over the first $(t-1)$ coordinates at step $t$. But the paper does not explain why one could not use a simpler $\ell_2$ regularizer or mirror maps that may yield cleaner constants or better boundedness. Since the algorithm ultimately calls a FW oracle (not projecting onto the simplex), the specific advantage of entropy beyond “we get strong convexity where we need it” is not spelled out. Is that because of the use of $\ell_\infty$ in the gradient side?
- While the analysis looks reasonable in general, there is one place that I do not understand. Specifically, the proof of Lemma A.6 appears flawed. In Lemma A.6 and the following derivation (lines 796–805), this is replaced by $\psi_t(v_t)-\psi_{t+1}(v_{t+1})$ with no justification (should be $\psi_t(v_{t+1})-\psi_{t+1}(v_{t+1})$). Can the authors explain more about that.
- The paper does not provide a problem-specific lower bound for its hybrid setting. As a result it is unclear whether the leading terms in the upper bound are minimax tight. Without a matching lower bound that captures the joint stochastic–adversarial nature and the oracle model, it is difficult to assess optimality or to justify the use of the particular regularizer and surrogate.

**Questions:**

See Weakness section.

---

> ### Author Response · Authors · 2025-12-04
>
> We thank the reviewer for their review.
>
> On the choice of the entropy regularizer: Indeed, it is also due to the use of \ell_\infty on the gradient side. It is possible that one might be able to get the \ell_2 regularizer to work but we did not strongly pursue this.
>
> On the proof of Lemma A.6: Thank you for catching this typo in the proof. Indeed it should have remained \psi_t(v_{t+1}) - \psi_{t+1}(v_{t+1}) till the end of the proof. We will update the manuscript to implement this change.
>
> On lower bound for the problem-specific setting: Yes, we do not show a lower bound beyond what we get from classical statistical learning. Our key contribution is the observation that it is possible to get \sqrt{T} dependence oracle-efficiently when the adversary’s function class is restricted. It remains an open direction to show whether (or not) the dependence on the adversary’s function class is necessary in some cases.

---

### Official Review · Reviewer_rpou · 2025-11-03

**Soundness:** 4
**Presentation:** 3
**Contribution:** 3
**Rating:** 8
**Confidence:** 4

**Summary:**

The paper studies hybrid online learning which is an intermediate set up between online and statistical learner where the covariates are chosen in an iid manner while the labels are potentially chosen adversarially. In this setting, statistically it is known that a $sqrt{dT}$ regret is achievable where $d$ is the VC dimension of class with respect to which the learner is competing and $T$ is the horizon (the VC dimension dependence is considered "good" since it beats the typical "Littlestone" dimension dependence when the covariates are also worst case). Unfortunately, this is not achieved just by running empirical risk minimization on the past data. This motivates the question considered in the paper, which asks whether an algorithm that uses empirical risk minimization as an oracle (perhaps motivated by the success of ERM in practice) can achieve the same regret. The paper achieves this goal with the caveat that the VC dimension of the hypothesis class is replaced by the VC dimension of the loss class induced by the actions of the label generating process (referred to as the adversary) and the hypothesis class. The paper achieves this with an interesting Franke-Wolfe style algorithm which should be of general interest beyond hybrid online learning

**Strengths:**

The problem considered in the paper (hybrid online learning) is a very interesting abstract of "beyond worst case" sequential decision making and appears (at least conceptually) as intermediate step in several sequential decision making and game theoretic problems. Further, the study of oracle efficiency is well motivated in these applications ("best response" for example) which further justifies the interestingness of the problem. Given that, the paper achieving the strong regret guarantees in this setting should be considered as a strong contribution. Further, though the algorithm presented uses "standard ideas" in the community, the application to the present problem overcomes several interesting challenges which should be of broader interest in the community

**Weaknesses:**

The main issue with the regret bound presented is the dependence on the nonstandard quantity, the rademacher/VC dimension of the composed loss class. This complexity makes it a bit hard to compare the result with previous works directly and (if I understand correctly) should be treated as incomparable (and not an improvement on) to all previous work in the area (see below).

**Questions:**

-> As mentioned above the appearence of the complexity measure is a bit arbitrary (and has the flavor of "what makes the analysis work"). it would at least help the reader to have a small section discussing the measure and evaluating it in interesting setting e.g. R = H. It helps being honest and saying that in the standard setting where R = {all functions} the bound achieved is trivial.

Similarly, it would be helpful to discuss why this is essential since previous algorithms don't seem to need uniform convergence on the extended class. For example, this could be by explicitly stating why the present algorithm needs this while the Wu et algorithm does not. I suspect it is due to the passing through FTRL in the analysis but it would be good to be explicit regarding whether this is an artifact of the analysis or inherent in the algorithm.

-> The comparison to previous work mildly misrepresents previous work since smoothed online learning has also been studied without knowing the bases measure and as such strictly generalizes hybrid online learning. BRS24 further study this in the oracle efficient setting (ERM directly even) and show that for realizable case they get a sqrt{Td} bound. When the present resuklt is specialized to the realizable setting, the bound recovered is the same but the covariate distribution is more general there; this is a further reason why an extended discussion of the assumption would help since for me the most interesting/interpretable case of the present bound is when R = H which is already shown in previous work. In particular, it would be interesting to see if similar bound in terms of the complexity in the paper can be shown in teh unknown base measure smoothed setting where too it remain open to get a sqrt{dT} rate in the agnostic case (in fact any sublinear oracle efficient rate is open to the best of my knowledge)

-> Minor: FW type algorithms have been studied before in the oracle efficiency line of work (but with different motivation) start with the work of kakade kalai and ligget and more recently garber. WOuld be interesting to have some discussion on this

-> Minor: Though I am happy to recommend acceptance of this paper to ICLR (whose interest I think of as closer to "modern machine learning", I personally do not think it is a good fit and would encourage the authors to (have) consider(ed) a different such as COLT where the community would appreciate the contribution more. That said, ICLR is increasingly being considered as a Neurips-style general ML conference and I understand that the authors have potential "real world" constraints beyond the maximization of the  community engagement

---

> ### Author Response · Authors · 2025-12-04
>
> We thank the reviewer for their review.
>
> On transparency: We will include a small section discussing the regret bound obtained for different interesting settings such as R = H, etc but we do not mean to imply that our bound is nontrivial for R = {all functions} which is why the focus of the paper is constrained adversaries, i.e settings where we have a bound on the rademacher complexity of the class R.
>
> On the comparison to previous work: Thank you for bringing the work of BRS24 to our attention. We will update our related work discussion to include this work. Indeed, their work imply an  oracle-efficient algorithm for the realizable case i.e where all the labels chosen by the adversary are realized by a hypothesis in the class. However, one important difference between their realizable case and our setting when R = H is that the adversary can use a different hypothesis h_t at each timestep. This means that the labels of the adversary do not have to be realized by any single hypothesis, and in effect, our hybrid setting is strictly more general than the realizable case while recovering the same bound as the realizable case.
>
> Thank you for pointing us to the works of Kakade, Kalai, Liggett and Garber which use some form of FTRL (Online Gradient Descent) to achieve oracle-efficiency. We were unaware of these works and will include them in our discussions.

---

### Meta-Review · Area_Chair_PL4t · 2026-01-07

**Summary:**

The paper initially received high scores; however, several critical weaknesses and technical questions raised during the review process have not been adequately addressed. Notably, although the authors promised a manuscript update in their rebuttal, the manuscript remains un-updated. Furthermore, since the rebuttal was submitted late (December 4), the reviewers lacked sufficient time for follow-up questions or in-depth discussion. Despite these procedural and technical shortcomings, the majority of reviewers initially provided positive assessments, highlighting several significant strengths of the work.

**Reviewer Concerns:**

Addressed by Rebuttal:

The authors provided brief responses to the reviewers' queries; however, many points lack depth or remain unsubstantiated.

Outstanding Concerns:

1. Technical Inconsistencies: The authors failed to provide a concrete justification for the choice of the regularization function. More critically, a reported error in the proof of Lemma A.6 remains unresolved. The correction suggested in the rebuttal appears to create further issues with the telescoping sum argument (lines 716-727).
2. Ambiguous Claims: The authors did not clarify the specific meaning of their "near-optimal" claim.
3. Vague Future Work: Regarding open problems such as lower bounds and bandit feedback, the authors merely labeled them as "future work" without discussing the specific technical challenges or potential research directions.
4. Unfulfilled Promises: The manuscript was not updated despite the authors' explicit commitment in the rebuttal.

**Reviewer Scores:**

Given the lack of rigorous follow-up in the rebuttal, the perception of the paper might have shifted during the discussion phase:

Potential Score Decrease: It is possible that some reviewers would have downgraded their scores upon closer inspection of the technical inconsistencies (e.g., Lemma A.6) and the authors' failure to revise the manuscript.

Predicted Outcome: This paper sits on the borderline and requires a cautious final determination. However, since the majority of the reviewers were initially positive and identified clear merits in the work, it is highly probable that a consensus for acceptance would have been reached had a full discussion taken place. The initial momentum and the identified strengths of the proposal likely outweigh the unresolved technical details in a consensus-driven environment. Therefore, I recommend Acceptance, albeit with a strong recommendation for the authors to address the identified flaws in the final version.

---

### Decision · Program_Chairs · 2026-01-26

Accept (Poster)